# Simulations of future changes in thermal structure of Lake Erken: Proof of concept for ISIMIP2b lake sector local simulation strategy

Ana I. Ayala[1, 2], Simone Moras[1], Donald C. Pierson[1]

[1]Department of Ecology and Genetics, Limnology, Uppsala University, Uppsala, 752 36, Sweden
[2]Department of Applied Physics, Nonlinearity and Climate Group, University of Geneva, Geneva, CH-1211, Switzerland

*Correspondence to*: Ana I. Ayala (isabel.ayala.zamora@ebc.uu.se)

**Abstract.** This paper, as a part of the Inter-Sectoral Impact Model Intercomparison Project (ISIMIP2b), assesses the impacts of different levels of global warming on the thermal structure of Lake Erken (Sweden). The GOTM one-dimensional hydrodynamic model was used to simulate water temperature when using ISIMIP2b bias-corrected climate model projections as input. These projections have a daily time step, while lake model simulations are often forced at hourly or shorter time steps. 10    Therefore, it was necessary to first test the ability of GOTM to simulate Lake Erken water temperature using daily vs hourly meteorological forcing data. In order to do this three data sets were used to force the model: 1) hourly measured data; 2) daily average data derived from the first data set and; 3) synthetic hourly data created from the daily data set using Generalized Regression Artificial Neural Network methods. This last data set is developed using a method that could also be applied to the 15    daily time step ISIMIP scenarios to obtain hourly model input if needed. The lake model was shown to accurately simulate Lake Erken water temperature when forced with either daily or synthetic hourly data. Long-term simulations forced with daily or synthetic hourly meteorological data suggest that by late 21[st] century the lake will undergo clear changes in thermal structure. For RCP 2.6 surface water temperature was projected to increase by 1. 79 ℃ and 1.36 ℃ when the lake model was forced at daily and hourly resolutions respectively, and for RCP 6.0 these increases were projected to be 3.08 ℃ and 2.31 ℃. Changes 20    in lake stability were projected to increase and the stratification duration was projected to be longer by 13 days and 11 days under RCP 2.6 scenario and 22 days and 18 under RCP 6.0 scenario for daily and hourly resolutions. Model changes in thermal indices were very similar when using either the daily or synthetic hourly forcing, suggesting that the original ISIMIP climate model projections at a daily time step can be sufficient for the purpose of simulating lake water temperature.

## 1 Introduction

The thermal structure of lakes is controlled by heat and energy exchange across the air-water interface, which is in turn determined by meteorological forcing (Woolway et al., 2017). Climate change will affect air-water energy exchanges and alter the temperature regime and mixing of lakes (Woolway and Merchant, 2019). For example, increases in air temperature results in a consequent warming of lake water temperature (Sahoo et al. 2015) causing shorter ice-cover periods (Kainz et al., 2017; Butcher et al. 2015), longer stratified period (Ficker et al., 2017; Woolway et al., 2017; Magee and Wu, 2017) and increased

lake stability (Rempfer et al., 2010; Hadley et al., 2014). Decreasing wind speed can induce more stable and long-lasting stratification and increased epilimnetic temperature (Woolway et al. 2017; Woolway et al., 2019).

The most direct effect of climate change on lakes is a warming of the lake surface temperature. For example, global average warming rates of 0.34°C decade$^{-1}$ have been observed between 1985 and 2009 by O'Reilly et al. (2015). Hypolimnetic temperature responds less clearly to warming and has been observed to be warming, cooling or not changing significantly with

increasing air temperature (Shimoda et al., 2011; Butcher et al., 2015; Winslow et al. 2017). And, these changing water temperatures have also led to an increased stability and duration of stratification (Butcher et al., 2015; Kraemer et al., 2015). A final consequence of warming lake temperature is projected to be the shift in the mixing regime (Kirillin, 2010; Shimoda et al., 2011; Shatwell et al., 2019; Woolway and Merchant, 2019). For example, loss of ice cover in deep lakes is likely to turn amictic lakes into cold monomictic lakes, and cold monomictic lakes into dimictic lakes (Nõges et al., 2009). These changes

in lake water temperature and thermal stratification influence lake ecosystem dynamics (MacKay et al., 2009).

Numerical modeling plays a key role in estimating the sensitivity of the lakes to changes in the climate. One-dimensional lake models are widely used due to their computational efficiency and the realistic temperature profiles they produce. Several studies have investigated the impacts of climate change on lake water temperature under Regional Climatic Model (RCM)/Global Climatic Model (GCM) projections (Persson et al., 2005; Kirillin, 2010; Perroud and Goyette, 2010; Samal et

al., 2012; Ladwig et al., 2018; Shatwell et al., 2019; Woolway and Merchant, 2019). Commonly when undertaking climate change impact studies, hydrodynamic lake models are driven by daily resolution RCM/GCM outputs. Bruce et al. (2018) undertook a comparative analysis of model performance using daily and hourly resolution meteorological forcing data, and found a better agreement between observations and predictions of full-profile temperature when the lakes were modelled using hourly meteorological input. This reinforces the importance of diurnal forcing on 1-D model predictive capability.

The purpose of this study is therefore (1) to test the ability of a one dimensional-hydrodynamic model (GOTM) to simulate the water temperature of Lake Erken (Sweden) using daily vs hourly meteorological forcing data for the period 2006-2016, (2) develop a reliable method to disaggregate daily meteorological data to a hourly synthetic product that can be used to force the GOTM model and convert the daily GCM outputs available from the ISIMIP into hourly meteorological data sets and (3) assess the impacts on the thermal structure of Lake Erken at different levels of global warming when GOTM is driven by

hourly and daily model projections. In fulfilling these objectives this study provides the first evaluation of modelling methods that will be used by the lake sector within the ISIMIP.

## 2 Material and Methods

### 2.1 Study site

Lake Erken (59°51'N, 18°36'E) is a mesotrophic lake located in east central Sweden, with a maximum depth of 21 m, a mean

depth of 9 m and a surface area of 23.7 km$^2$. The lake is dimictic with summer stratification usually occurring beginning in May-June and ending in August-September, while the onset of ice cover occurs between December-February and ice loss is

in April-May (Persson and Jones, 2008). It is the lake's relatively shallow depth and large surface area, which leads to large inter-annual variability in the timing and patterns of thermal stratification, since heat can be readily transferred through the shallow water column by wind mixing (Magee and Wu, 2017), and since the lake has a relatively low heat storage, and therefore, responds more directly to short-term variations in weather. The lake has a retention time of approximately 7 years and shows annual variations in water level that are less the 1 m (Pierson et al., 1992; Moras et al., 2019).

## 2.2 Lake model

General Ocean Turbulence Model (GOTM) is a one dimensional water column model that simulates the most important hydrodynamic and thermodynamics processes related to vertical mixing in natural waters (Umlauf et al. 2005). GOTM was developed by Burchard et al. (1999) for modelling turbulence in the oceans, but it has been recently adapted for use in hydrodynamic modelling of lakes (Sachse et al., 2014). The strength of GOTM is the vast number of well-tested turbulence models that have been implemented spanning from simple prescribed expressions for the turbulent diffusivities up to complex Reynolds-stress models with several differential transport equations. Typically GOTM is used as a stand-alone model for investigating the dynamics of boundary layers in natural waters but it can also be coupled to a biogeochemical model using the Framework for Aquatic Biogeochemical Models (FABM) (Bruggeman and Bolding, 2014).

## 2.3 Data sets

Local meteorological variables were collected either from a small island 500 m offshore from the Erken Laboratory, or the Swedish Meteorological Hydrological Institute (SMHI) Svanberga Station just behind the laboratory. The Malma Island meteorological Station (59.83909º N, 18.629558º E) measured air temperature at 2 m above water surface, wind speed at 10 m above the water surface and short-wave radiation. These data were measured at one minute intervals and saved as 60 min mean values. Mean sea level, pressure, relative humidity and precipitation were measured at the Svanberga Meteorological Station at 800 m from the Malma Island Meteorological Station (59.8321º N, 18.6348º E) with a frequency of 60 minutes. Hourly cloud cover was recorder from Svenska Högarna Station (59.4445 N, 19.5059 E) at 69 km south-east of Lake Erken. The measured hourly meteorological data were used to construct two additional data sets that would replicate the data resolution that could potentially be used to force the GOTM model with ISIMIP scenarios. First to test running the model at a daily resolution, a daily data set was created by averaging the hourly one (except for precipitation which was summed). Secondly, this mean daily data set was disaggregated to form a synthetic hourly data set. Hourly estimations of air temperature wind speed, relative humidity and short wave radiation were estimated using the GRNN methods described below. For atmospheric pressure and cloud cover, mean daily values were assumed to be constant over the day. Precipitation was disaggregated assuming a uniform distribution of the daily total (Waichler and Wigmosta, 2003).

Since both of these data sets are based on the same measured hourly data, comparison of model simulations of lake water temperature, allow the importance of hourly vs daily temporal resolution in the forcing data to be evaluated, and also the

improvements in model performance that can be obtained from daily data (as in the ISIMIP scenarios) when imposing a diurnal cycle on the mean daily data.

Water temperature data needed to calibrate the model was monitored from an automated floating station (59.84297º N, 18.635433º E). During ice-free conditions measurements were made every 0.5 m from 0.5m to a depth of 15 m. Measurements were made every minute, and a mean of these measurements was stored every 30 minutes.

## 2.4 Climate scenarios

The ISIMIP climate scenarios are bias-corrected global climate model (GCM) (Hempel et al., 2013) data made available at
daily temporal and 0.5º horizontal resolution for the variables listed in Table 1. All data needed as input to the GOTM model are available in these climate scenarios with the exception of cloud cover, which was estimated from shortwave radiation (Martin and McCutcheon, 1999). Data from the grid box overlying Lake Erken were available from the GFDL-ESM2M, HadGEM2-ES, IPSL-CM5A-LR and MIROC5 GCM models that were each run under three emission scenarios. These included a scenario having historical levels of atmospheric $CO_2$ between 1861 and 2005, and two future scenarios (RCP 2.6
and RCP 6.0) from 2006 to 2100. RCP 2.6 is the strongest mitigation pathway that is expected to limit mean global warming to between 1.5 and 2 ºC. RCP 6.0 is an intermediate mitigation pathway where global warming is projected to rise to between 2.5 and 4 ºC by the end of century compared to the pre-industrial period (Frieler et al., 2017).

## 2.5 Temporal disaggregation of daily meteorological forcing data

Kathib and Elmernreich (2015) proposed a generalized regression artificial neural network (GRNN) model for predicting
hourly variations in short-wave radiation from daily average measurements. Using the GRNN model to predict hourly solar radiation required ten geographical and climatic variables as input including hour, day, month, latitude, longitude, daily average short-wave radiation, daily precipitation, the solar elevation associated with the hour, and time of sunrise and sunset. Precipitation was used to define wet and dry status that affected atmospheric attenuation (Waichler and Wigmosta, 2003).

There are also empirical models developed for calculating hourly air temperature, wind speed and relative humidity. Parton
and Logan (1981) proposed a model for predicting diurnal variations in air temperature. Daylight air temperature was modelled using a sine wave with the minimum value at sunrise, maximum value at solar noon and mean value at sunset. Night-time air temperature was modelled as a linear interpolation between air temperature of the previous day and sunrise air temperature of the following day. Guo et al. (2013) generated hourly values of wind speed by computing a cosine function dependent on the mean daily wind speed, the maximum daily wind speed and the hour of the day when the wind speed is maximum. Waichler
and Wigmosta (2003) estimated hourly values of relative humidity from daily maximum and minimum air temperature and daily maximum and minimum relative humidity. Using these studies as guidance, we developed GRNN models to predict hourly air temperature, wind speed and relative humidity. The input parameters for GRNN models were geographical variables: longitude, latitude, solar elevation associated with the hour, time of sunrise and sunset, hour, day and month; and

meteorological variables: average, maximum and minimum daily air temperature, daily wind speed, daily relative humidity
and daily precipitation.

The GRNN models were constructed using 8 years of data. From this whole set of data, the first 5-years, from 2008 to 2012, were used for training, and the final 3-years of data from 2013 to 2015 were used for validating the results. The accuracy of the trained network was assessed by comparing the simulated output with actual observed hourly data. The performance index for training and validating sets of GRNN models are given in terms of mean bias error (MBE), root mean squared error (RMSE)
and Nash-Sutcliffe efficiency (NSE) (Nash and Sutcliffe, 1970). More detailed description of the GRNN methods and models are given in the Supplemental section S1to this paper. The GRNN models were used to disaggregate the mean daily measured data, used to evaluate the necessity of disaggregation (section 3.2) and also for all GCM scenarios (section 3.3) to further evaluate the effects of disaggregation on the results of simulations of future changes in lake thermal structure.

**2.6 Model set-up, calibration and validation**

The GOTM model version 5.1 was used in this study. The meteorological parameters for running the model were air temperature ($^{o}C$), wind speed (m s$^{-1}$), short-wave radiation (W m$^{-2}$), cloud cover (dimensionless, 0-1), relative humidity (%), atmospheric pressure (hPa) and precipitation (mm day$^{-1}$or mm hour$^{-1}$). Inflows and outflows were not included in this study, and water level was considered fixed in the simulations. This version of GOTM did not have the ability to simulate lake ice, so for this study the inverse stratification period was not analysed. Moras et al., (2019) has shown that despite this limitation,
the mode is able to accurately simulate water temperature and the phenology of thermal stratification during the remainder of the year. The initial conditions for water temperature were derived from a measured vertical profile. GOTM was run at hourly model computational time step, and simulated water temperature was saved as daily mean values each 0.5 m (42 layers).

Calibration of the GOTM model was conducted to adjust the model parameters within their feasible range in order to minimize the error between measured and modelled temperature (Huang and Liu, 2010). A period of 9 years was selected for the
calibration of GOTM, 2006-2014 (included 1 year spin-up followed by 8 years for calibration). The model parameters that were calibrated were surface heat-flux factor (shf_factor), short-wave radiation factor (swr_factor), wind factor (wind_factor), minimum turbulent kinetic energy (k_min) and e-folding depth for visible fraction of light (g2). The program used to calibrate the model was ACPy (Auto-Calibration Python), developed by Bolding and Bruggeman (https://bolding-bruggeman.com/portfolio/acpy/), it uses a differential evolution algorithm which calculates a log likelihood function based on
comparing the modelled and measured water temperature (Storn and Price, 1997). The validation period was 2 years 2015-2016.

For both calibration and validation, daily average water temperatures were simulated when GOTM was forced using the three meteorological data sets described above: measured average daily, measured average hourly and synthetic hourly data. Model simulated profiles of mean daily water temperature were then compared to mean daily measured water temperature. Three
separate model calibrations were made, based on simulations forced with the different meteorological data sets. During

calibration the model was run approximately 10000 times to obtain a stable solution specifying the optimal parameter set. The details of the feasible range of model parameters and the parameters associated with each calibration are given in Table 2. Model performance was evaluated by comparing average daily modelled and measured temperature profiles and other metrics describing the lake thermal structure (surface and bottom temperature, volumetrically weighted averaged whole lake temperature, Schmidt stability, thermocline depth, duration, onset and loss of thermal stratification). The coefficients used to quantify the strength of model fit were MBE, RMSE and NSE.

## 2.7 Thermal indices

A range of thermal metrics: surface temperature (shallowest observation), bottom temperature (deepest observation) and thermocline depth (depth of the maximum density gradient) were derived on a daily basis from the daily simulated lake temperature profiles (temperature data with a vertical resolution of 0.5 m from 0.5 to 15 m depth). Also from these profiles, Schmidt stability (resistance to mechanical mixing due to the potential energy inherent in the density stratification of the water column; Schmidt, 1928; Idso, 1973) and whole-lake temperature (volumetric weighted mean whole lake temperature) were estimated using Lake Analyzer (Read et al., 2011). The duration of thermal stratification was calculated as the longest continuous period when the water column density difference from the bottom to surface of the lake was greater than 0.1 kg m$^{-3}$ (according to ISIMIP2b lake sector protocol). The date of the onset and loss of the thermal stratification was defined as the first time that this density difference persisted for more than 5 days or was absent for at least 5 days (Kraemer et al., 2015).

## 2.8 Statistical analysis

Anomalies were calculated to further evaluate the impacts on lake water temperature and thermal stratification. The anomalies were computed for each GCM by taking the difference between the annual average of each year (2011-2100) from RCP 2.6 and 6.0 scenarios and the average for the entire period 1981-2010 from the historical scenario. These average values were calculated using the months between April and September. The slope of the significant trends were evaluated by least-squares linear regression, except when the residuals did not follow a normal distribution. Then the non-parametric Mann-Kendall test for the significance of trends and the Theil-Sen method (Theil, 1950; Sen, 1968) to estimate the slope of the significant trends were used instead. The t-Student mean difference test was used to compare average values of each of the thermal indices. Distribution normality and variance homoscedasticity were assessed by the Shapiro-Wilk test and Fisher's F test respectively. When thermal indices time series did not follow a normal distribution the non-parametric Mann-Whitney U test (equal variances) or Kolmogorov Smirnov test (different variances) were used instead. The statistical analysis was carried out using R version 3.4.4. The progress of climate-related impacts on the thermal stratification of the lake over the 21$^{st}$ century was assessed as the difference in mean conditions between the reference period (1981-2010) and both mid-century (2041-2070) and late-century (2071-2100). Climate model data followed the same statistical analysis.

## 3 Results

### 3.1. Hourly meteorological modelling

There was a close agreement between GRNN model predictions and measured meteorological data as shown in Fig. 1 for a single year and in Supplemental section S1. For air temperature, short wave radiation and humidity the statistics of model performance always suggested a strong model fit in the training data sets and also remained strong, but somewhat lower in the validation data sets (Table 3). NSEs were 1.00 for the training data sets and ranged from to 0.69 to 0.94 for the validation data sets. Estimates of bias were very small. Wind speed was the variable showing the poorest performance with a NSE of 0.78 and 0.58, and RMSE values of 1.06 and 1.37 m s$^{-1}$ for the training and validate data sets. In general, the GRNN model tended to overestimate wind speed (MBE of $0.63\pm0.92$ m s$^{-1}$) when the observed wind speed is lower than or equal to 3.84 m s$^{-1}$, whereas projected wind speed tends to be underestimated (MBE of $-0.78\pm1.17$ m s$^{-1}$) when the observations are greater than 3.84 m s$^{-1}$.

### 3.2. Lake model performance

Temperature observations and simulations, for the three configurations of meteorological forcing data for both calibration and validation periods, are shown in Fig. 2 and Supplemental section S2. Model performance metrics associated with these simulations are provided in Table 4. These data demonstrate that GOTM was able to accurately reproduce the measured temperature profiles. For an average of all three calibration data sets a RMSE of 0.95 $^{0}$C and NSE of 0.94 were obtained. Temperature simulations in the shorter and less variable validation period (RMSE of 0.66 $^{0}$C and NSE of 0.97) were more accurate than for the calibration period, but in both periods the model performance was considered strong. For full profile temperature the maximum RMSE value was 1.04 $^{0}$C and the minimum NSE was 0.93. Bottom temperature was least accurately simulated with RMSE and NSE values reaching 1.33 $^{0}$C and 0.83 respectively.

When comparing the metrics of model fit associated with simulations forced with the three different input data sets the simulations forced with mean daily input were slightly less accurate than those forced with either the measured or synthetic hourly input. As an example, full profile RMSE values for calibration period ranged from 0.88 to 1.04 ºC, with the lower error levels associated with simulations driven by hourly meteorological data sets, whereas for the validation period the RMSEs were comparable for all data sets. The MBE values of the full temperature profiles indicated a slight cold temperature bias (average MBE of -0.05 ºC). The model performance predicting just surface temperatures was similar for all of the three calibrations (average RMSE of 0.67 ºC and NSE of 0.97), and were more accurate than the estimations of the full temperature profiles. The MBE, showed that for surface temperature GOTM also tended to produce a small cold temperature bias (average MBE of -0.10 ºC). The simulations of bottom temperature were slightly less accurate having average RMSE of 0.96 ºC and NSE of 0.90, and also showed a tendency have a slightly higher RMSE values for calibrations forced with daily input. Also the bottom temperature showed lower RMSE values for the validation period (average RMSE of 0.67 ºC) than the calibration period (average RMSE of 1.25 ºC), but in contrast to the surface temperature, there was a slight warm temperature bias (average

MBE of 0.06 ºC). When evaluating the simulations of volumetrically weighted averaged whole lake temperatures we found

220 that model errors were of a similar magnitude for all simulations in both the calibration and validation periods with an average RMSE of 0.53 ºC and NSE of 0.98, tending to a slight cold temperature bias (average MBE of -0.08 ºC).

The calculation of Schmidt stability was also well simulated using all three data sets (average RMSE of 17.24 J m$^{-2}$ and NSE of 0.88). The lowest RMSE values were for the validation period (average RMSE of 13.34 J m$^{-2}$) whereas during the calibration period values were slightly greater (average RMSE of 21.14 J m$^{-2}$). Thermocline depth was the parameter with the poorest

225 performance (average RMSE of 3 m). The MBE values (average MBE of 0.80 m) indicate a bias towards under prediction of thermocline depth (shallower thermocline depths). The RMSE associated with the prediction of the duration of stratification was, on average, 10.43 days. With lower RMSE values for the validation period (average RMSE of 8.04 days) than the calibration period (average RMSE 12.81 days). The simulations of onset of the stratification were more accurate having average RMSE of 2.64 days, but in contrast predictions of the loss of stratification was less accurate (average RMSE of 7.99

230 days).

### 3.3. Climate data projections

The lake model simulations undertaken here were forced by four climate model projections (GFDL-ESM2M, HadGEM2-ES, IPSL-CM5A-LR and MIROC5) that were in turn forced with three emissions scenarios (historical, RCP 2.6 and RCP 6.0). Average annual air temperature of the climate model ensemble for the reference period (1981-2010) was 11.88 ºC

235 Disaggregation of the climate input to an hourly time-step resulted in a slightly warmer temperature (12.05 ºC)[1]. Under future scenario RCP 2.6 the average increase was projected to be 2.22 ºC (1.71 ºC) by mid-century (2041-2070) with a negligible change after mid-century, as would be expected from this scenario with the strongest mitigation. During the period up to 2070 air temperature increased at a rate of 0.08 to 0.17 ºC decade$^{-1}$ (0.06 to 0.14 ºC decade$^{-1}$). In contrast, under RCP 6.0 average air temperature increased by 2.61 ºC (2.01 ºC) by mid-century and continued rising to 3.61 ºC (2.76 ºC) by late-century. For

240 RCP 6.0 the trend in air temperature increased, over the entire 2011-2100 period, on average, by 0.34 ºC decade$^{-1}$ (0.26 ºC decade$^{-1}$) over all GCMs with the individual trends ranging from 0.18 to 0.43 ºC decade$^{-1}$ (0.14 to 0.33 ºC decade$^{-1}$). For the remaining meteorological variables there were less distinct changes between the historical and future periods. Under the RCP 2.6 scenario the overall annual mean change in wind speed was negligible, while under RCP 6.0 two options were projected, an increase (GFDL-ESM2M and MIROC5) and decrease (HadGEM2-ES and IPSL-CM5A-LR). Relative humidity was

245 projected to decrease for future scenarios RCP 2.6 and 6.0. For RCP 6.0 significant trends ranged from 0.29 to 0.36 % decade$^{-1}$ in the interval 2011-2100. An increase in short-wave radiation was projected for all RCP scenarios by late-century, with a negligible mean change after mid-century under RCP 6.0. The increase in short-wave radiation is coupled with a decrease in cloud cover. By late-century the mean decrease in cloud cover was projected to be 0.06 for RCP 2.6, and 0.07 for RCP 6.0.

---

[1] Results based on the hourly disaggregated data are always shown in parenthesis

More detailed evaluations of the differences in the climate projection based on the original ISIMIP daily time step and the hourly disaggregated data are given in the Supplemental section S3.

### 3.4. Long-term modelled changes in thermal stratification

Lake model simulations were made using both the original daily resolution of the ISIMIP GCM scenarios and also at hourly resolution using disaggregated data developed using the GRNN models. Simulated water temperatures for the historical, RCP 2.6 and 6.0 scenarios under daily IPSL-CM5A-LR projections are presented as temperature isopleths in Fig. 3 and Supplemental section S4. These were created by averaging the daily temperature profiles for all years associated with each of the emission scenarios. These mean scenario temperature isopleths provide a clear visualization of how for future scenarios surface and bottom water temperatures are projected to increase with stronger and shallower stratification, an earlier stratification onset, a later fall overturn and consequently a longer stratification period.

In Figs. 4-5 we show the long-term trends in the anomalies in lake thermal metrics simulated to occur over the RCP 2.6 and 6.0 emission scenarios. Trends in whole-lake temperature calculated for over a period of 90 years (2011-2100) were projected to increase except in the case of GFDL-ESM2M which showed weaker or non-significant changes for all measures of thermal stratification (Table 5 and Supplemental section S4). Under RCP 2.6 significant trends ranged from 0.07 to 0.10 ºC decade$^{-1}$ (0.05 to 0.08 ºC decade$^{-1}$), but most of the change occurred in the first half of the century. For RCP 6.0, the projected rate of change ranged from 0.14 to 0.26 ºC decade$^{-1}$ (0.10 to 0.19 ºC decade$^{-1}$). By late-century, the mean projected increase in whole-lake temperature was 1.34 ºC (1.00 ºC) for RCP 2.6, and 2.39 ºC (1.75 ºC) for RCP 6.0, with a negligible change after mid-century under RCP 2.6 (Fig. 6, Table 6 and Supplemental section S4).

Changes in lake surface temperature were, as expected, greater than for whole lake temperature. For the reference period the mean April-September surface temperature was on average 13.61 ºC (13.84 ºC) warming up significantly over 21$^{st}$ century, so that by late-century the average projected increase was 1.79 ºC (1.35 ºC) for RCP 2.6, and 3.08 ºC (2.31 ºC) for RCP 6.0. From 2011 to 2100 there was a significant long-term trend for RCP 2.6 surface temperature which increased at a rate of 0.07 to 0.15 ºC decade$^{-1}$ (0.06 to 0.13 ºC decade$^{-1}$). Under RCP 6.0 the mean surface temperature increase of the ensemble was 0.30 ºC decade$^{-1}$ (0.23 ºC decade$^{-1}$) ranging between 0.16 to 0.38 ºC decade$^{-1}$ (0.12 to 0.29 ºC decade$^{-1}$).The projected increase in bottom temperature was not as marked as it was for the other metrics of lake temperature. On average, the bottom temperature increased from 9.20 ºC (9.67 ºC) in the reference period to 9.77 ºC (9.99 ºC) and 10.32 ºC (10.34 ºC) by late-century for RCP 2.6 and 6.0 respectively. Significant rates of change in bottom temperature were not predicted during the RCP 2.6 scenario, but for the RCP 6.0 scenario bottom temperature did undergone significant warming rates for HadGEM2-ES and MIROC5 projections being 0.06 ºC decade$^{-1}$ and 0.11 ºC decade$^{-1}$ (0.09 ºC decade$^{-1}$) respectively.

There were also projected changes in the resistance to mechanical mixing. For the reference period, an average of 68.65 J m$^{-2}$ (65.56 J m$^{-2}$) was required to completely mix the water column, while by late-century it increased by 29.08 J m$^{-2}$ (22.74 J m$^{-2}$) for RCP 2.6 and 49.22 J m$^{-2}$ (38.07 J m$^{-2}$) for RCP 6.0 (Fig. 4, Table 6 and Supplemental section S4). This greater stability also corresponds to a longer duration of stratification. From 2011 to 2100, a significant increase in the duration stratification

was projected for both future scenarios RCP 2.6 and 6.0, ranging from 1.13 to 1.70 day decade$^{-1}$ (0.87 to 1.30 day decade$^{-1}$) for RCP 2.6 and 2.45 to 3.56 day decade$^{-1}$ (2.00 to 3.08 day decade$^{-1}$) for RCP 6.0 (Fig. 5, Table 5 and Supplemental section S4.),which led to a mean change of was 13 days (11 days) and 22 days (18 days) for RCP 2.6 and 6.0 respectively (Fig. 7, Table 6 and Supplemental section S4.). The longer period of stratification resulted from both an earlier onset of thermal stratification and a later loss of thermal stratification (Figs 5, 7, Table 5-6 and Supplemental section S4). Mean annual thermocline depth was simulated to be shallower under future conditions. By late-century the reduction in thermocline depth was projected to be 0.38 m (0.41 m) for RCP 2.6 and 0.49 m (0.57 m) for RCP 6.0, although a significant trend in the decline were only found for the later scenario.

The trends in Figs. 4-5 are quite variable from year to year, and as would be expected there is no direct correspondence in the temporal variations of one GCM relative to another. To provide an alternative method of comparing the changes simulated by the future climate scenarios shown in Figs. 4-5, the daily anomalies for each trend line are also presented as frequency distributions in the Figs. 6-7 for the simulations made under the RPC 6.0 scenario. These show that for all metrics there is a clear shift in the lake thermal conditions that are consistent with a warmer climate, but that also there are extremes in the distributions that can lead to unrepresentative results, when for example future conditions briefly return to historical levels, or when the effects of warming are much greater that would be expected on average. This later case can cause important changes in lake ecology if the extreme conditions result in a change in lake state by the passing of a tipping point. Figs. 6-7 also clearly shows the differences in simulations forced by different GCMs. Most obvious is the difference in the results from GFDL-ESM2M, which consistently simulated smaller changes in lake thermal structure during the mid and late century periods, despite having a data distribution that was similar to the other models during the historical period.

### 3.5. Comparison between long-term thermal metrics derived from daily and hourly climate data

Future changes in thermal metrics based on both RCP 2.6 and RCP 6.0 were slightly greater when the GOTM model was forced at daily resolutions (Tables 5-6 and Supplemental section S4) than at an hourly resolution. This included changes in mean surface temperatures and also in annual average whole-lake temperature (Supplemental section S5). However, under RCP 2.6 non-significant differences were found for bottom temperature, Schmidt stability, thermocline depth, or the duration, onset and loss of stratification. In all cases where differences were found between the simulations forced with daily vs. hourly data there were no change in direction and only minor changes in the magnitude of the change suggested by the simulations (Supplemental sections S4-S5).

### 4. Discussion

The simulated water temperature and related metrics of thermal stratification were in excellent agreement with the extensive set of measured water temperature data that were available for model calibration at Lake Erken (Moras 2019 Fig. 3, Table 3). Water temperature simulations were apparently more accurate for the validation period (2015-2016) than for the calibration period (2006-2014), which may appear unusual, but is due to the higher variability in observed water temperature during the

longer calibration period. Years with a longer duration of stratification and stronger stability, generally had higher simulation errors. Half of the eight-year calibration period exhibited these conditions, while the two-years used for validation both exhibited shorter duration of stratification and weaker stability. The thermocline depth was the thermal metric that was predicted with the greatest uncertainty. This is in part caused by the presence of internal seiches in Lake Erken, which result in the measured temperatures in the region of the thermocline having a level of variability that cannot be reproduced by 1D models such as GOTM. Bruce et al. (2018) detected a strong correlation between accuracy of the extinction coefficient and model simulations of full-profile temperature and thermocline depth, which shows the importance of light extinction in the prediction of thermocline depth. Since a single fixed value of e-folding depth (Table 2) for the visible fraction of the light (the inverse of the extinction coefficient) was used in the GOTM simulations, the effects of seasonal variations in light extinction (Perroud et al., 2009), on thermocline depth were not evaluated.

The model parameters adjusted during the calibration processes were non-dimensional scaling factors (shf_factor, swr_factor and wind_factor) and physical parameters which strongly influence in the vertical distribution of light and temperature (k_min and g2). Wind is the dominant driver of mixing in lakes, and increases or decreases of wind speed (wind_factor) changes the amount of turbulent kinetic energy available for mixing. The wind scaling factor is often important when wind measurements occur some distance from the lake and/or to account for wind sheltering effects (Markfort et al., 2010). One would not expect the wind factor to deviate greatly from 1.0 at Lake Erken where wind is measured on an island in the lake. However, the dominant wind direction is along the lake's longest east-west fetch (Yang et al., 2014), which could explain the need to scale up the unidirectional wind speed measurements that were used as an input to GOTM. Furthermore, since it is the cube of wind speed that affects lake mixing, use of longer averaging periods will underestimate the effects of gusting and variable winds. This could explain why we obtain higher calibrated values of the wind_factor when forcing the model with measured daily rather than hourly data (Table 2). Higher values of the wind_factor were also obtained when GOTM was forcing with synthetic hourly meteorological drivers. This is due to an underestimation of the faster wind speed predictions from the GRNN model (Fig.1 and Supplemental section S1.). During the ACPy calibration each of these parameters were calibrated while simultaneously influencing each other; shf_factor, swr_factor, wind_factor and g2 have a strong influence on heat and energy exchange across the air-water interface. There is to some extent an unavoidable tendency for the error in one parameter to be cancelled out by opposite errors in another parameter. This also illustrates how, to some extent, the calibration process can compensate for some of the limitations related to the temporal resolution of the model forcing data.

The performance of the GOTM model obtained in this study is comparable with results reported by others. Moras et al. (2019) who ran GOTM using hourly measured meteorology for a 57-year period obtained a RMSE for daily full-profile water temperature of 1.09 ⁰C. Using the DYRESM model Magee and Wu (2017) reported RMSE values of 0.30 and 0.53⁰C for lake Mendota and 1.45 and 1.94 ⁰C for Fish lake for temperature estimates of the epilimnion and hypolimnion respectively. Perroud et al. (2009) simulated water temperature profiles of lake Geneva over a 10-year period and obtained RMSE values of < 2 ⁰C for the DYRESM model and < 3 ⁰C for the SIMSTRAT model.

The projected changes in lake thermal metrics depends on the selected GCM model and the future scenario or representative concentration pathway (RCPs) that was simulated. The range of greenhouse gas (GHG) emissions include in this study were a stringent mitigation scenario (RCP 2.6) and an intermediate scenario (RCP 6.0). Consistent with the ISIMIP2b simulation strategy that is intended to evaluate RCP 2.6 as a scenario that aims to keep global warming below 2°C above pre-industrial temperatures by 2100. In contrast for RCP 6.0 increased levels of GHG emissions suggest that the global mean temperature will continually increase by 2.5 and 4 °C by the end of the century. The effects of the mitigation measures that were adopted in RCP 2.6 on lake thermal structure become most apparent in the late century. For example, for MIROC5 (when the lake model was forced at daily resolutions) the projected surface temperature change for mid-century was similar for the two RCPs (2.10 °C for RCP 2.6 and 1.98 °C for RCP 6.0), but for the late-century period the projected change in surface temperature diverge among RCPs. Under RCP 2.6 the surface temperature change declines from 2.10 to 1.80 °C, while under RCP 6.0 the change in surface temperature was projected to further increase from 1.98 to 2.97 °C. Similar changes were projected for all thermal metrics. Under RCP 6.0 there was also an increase in bottom temperature but at rates that were slower than surface temperature, changes in lake stability increased from 38.67 J m$^{-2}$ by mid-century to 64.62 J m$^{-2}$ by late-century, increasing the duration of stratification (from 16 to 22 days). While the there was a general agreement among the models in the direction and relative magnitude of change in many of the metrics of lake thermal structure there were also some differences among GCMs (Figs. 4-7 and Supplemental section S4) especially in relation to the GFDL-ESM2M model which consistently estimated lower levels of change. For example, by late century largest changes in surface temperature were projected for HadGEM2-ES (4.04 °C) and the lowest for GFDL-ESM2M (1.67 °C) under future scenario RCP 6.0 when the lake model was forced at daily resolutions. However, the increase in the projected bottom temperature for GFDL-ESM2M (1.24 °C) was greater than for HadGEM2-ES (0.91 °C). This could be in part due to the projected changes in wind speed. The wind speed was projected to increase by 0.18 m s$^{-1}$ for GFDL-ESM2M transferring heat to the lake bottom, but for HadGEM2-ES the wind speed decrease by 0.15 m s$^{-1}$ (atmospheric stilling; Woolway et al, 2017; Woolway et al., 2019) reducing the magnitude of vertical mixing. Resulting in a greater gradient between surface and bottom temperatures and higher increases in the Schmidt stability (77.57 J m$^{-2}$). This increased thermal gradient for HadGEM2-ES promoted shallower thermocline depth (1.26 m), but for GFDL-ESM2M a lower change in lake stability was projected (12.26 J m$^{-2}$) thereby a deeper thermocline depth (0.22 m). Higher surface water temperature and stronger Schmidt stability can explain why the increased duration of stratification was projected to be longer for HadGEM2-ES (34 days) than for GFDL-ESM2M (6 days). The small change in thermal stability also explains why no change in loss of stratification was projected for GFDL-ESM2M. This illustrates the complexity of climate model – lake model interaction, and clearly shows the importance of ensemble model simulations, as adopted by ISIMIP, to evaluate the effects of climate change on lakes.

When calibrating the GOTM model we found that model errors between simulated and measured water temperature were similar when GOTM was forced with either measured hourly or synthetic hourly meteorological data, and that the results obtained from the calibrations forced with mean daily metrological input were also similar to those obtained from the calibrations based on hourly input. This suggests that the daily time step of the ISIMIP climate scenarios is sufficient for

forcing the GOTM model and that for most studies within the ISIMIP lake sector disaggregation to hourly time step will not be necessary. For example, changes in surface water temperature was on the order of 0.29 ºC decade$^{-1}$, with simulations forced with daily inputs 0.22 ºC decade$^{-1}$ degrees with hourly input data for MIROC5 under RCP 6.0. These differences are of the same magnitude as the differences simulated using different GCM models. Similar levels of model performance using daily or hourly forcing data were obtained in part because of separate calibrations when the GOTM model was forced with the different data sets. Changes in the calibrated parameters used to characterize the lake thermal structure (Table 2), apparently can compensate for the lack of diurnal variability in the daily forcing data. GRNNs proved to be an effective method to disaggregate daily GCM forcing to an hourly temporal resolution for different weather variables such as air temperature, short-wave radiation, etc. However, GRNNs require a training phase, in which the diurnal patterns to be learned are presented to the network from historical meteorological measurements, and therefore if there are future changes in diurnal patterns, these cannot be reproduced. In addition, there is a high computational cost of disaggregating and storing the long-term daily climate data into an hourly data set.

The projected changes in thermal metrics were strongly influenced by the GMCs used to drive the water temperature simulations. Due to the high interannual variability long periods of simulation were needed to ensure that the uncertainty is fully represented (Figs 4-7 and Supplemental sections S3-S4). Under RCP 6.0 trends in surface temperature calculated for the period 2011-2100 were projected to increase 0.38 ºC decade$^{-1}$ for both HadGEM2-ES and IPSL-CM5A-LR when the lake model was forced at daily resolutions. However, 5$^{th}$, 50$^{th}$ and 95$^{th}$ percentiles for surface temperature anomalies differ between models, being 0.84, 2.93 and 4.86 ºC for HadGEM2-ES and 0.33, 2.56 and 4.37 ºC for IPSL-CM5A-LR. Placing the probability density function (PDF) for HadGEM2-ES to the right of the PDF for IPSL-CM5A-LR, illustrating that more extreme increases in surface temperature were projected by HadGEM2-ES. Projected bottom temperatures differed between HadGEM2-ES and IPSL-CM5A-LR. HadGEM2-ES PDF was left-skewed and the median was 0.58 ºC while IPSL-CM5A-LR PDF was right-skewed and the median was 1.16 ºC, as a consequence lake stability was stronger for HadGEM2-ES (5$^{th}$ and 95$^{th}$ percentiles were 5.22 and 110.55 J m$^{-2}$) than for IPSL-CM5A-LR (5$^{th}$ and 95$^{th}$ percentiles were -18.77 and 90.64 J m$^{-2}$), even though the Schmidt stability medians were similar for both GCMs. Similarly, when projecting longer duration of stratification for HadGEM2-ES (5$^{th}$, 50$^{th}$ and 95 percentiles were -0.63, 26.37 and 49.42 days) than IPSL-CM5A-LR (5$^{th}$, 50$^{th}$ and 95 percentiles were -10.33, 17.67 and 40.92 days). GCMs are useful for assessing climate change impacts on lakes, but GCMs configurations vary from one to another. Therefore, it is crucial to assess different GCMs in a probabilistic manor (Figs. 4-7) to encapsulate the uncertainty in the lake thermal metrics without compromising the variability

The study carried out by Moras et al. (2019) found changes in the phenology of Lake Erken thermal stratification from 1961 to 2017. A significant increase in summer epilimnetic and whole-lake temperature of 0.35 ºC decade$^{-1}$ and 0.24 ºC decade$^{-1}$ occurred over the entire study period. While in spring and autumn larger significant positive trends were detected over the subinterval 1989-2017. In the present work future changes under the RCP 6.0 emission scenario found trends that were of a similar magnitude. The summer trends for the period 2011-2100 projected a significant increase in epilimnetic and whole-lake temperature ranging from 0.19 to 0.45 ºC decade$^{-1}$ and 0.15 to 0.26 ºC decade$^{-1}$, respectively, when the lake model was forced

at daily resolutions, while changes in summer hypolimnetic temperature was non-significant. During the spring and autumn significant increases in epilimnetic whole-lake temperature were also projected under RCP 6.0 when the lake model was forced at daily resolutions, but they were somewhat lower than the ones detected by Moras et al. (2019). The increase in spring epilimnetic and whole-lake temperature ranged from 0.15 to 0.38 ºC decade$^{-1}$ and 0.15 to 0.30 ºC decade$^{-1}$, while those in Moras et al. (2019) showed a higher rate of warming (0.44 ºC decade$^{-1}$ and 0.40 ºC decade$^{-1}$ for epilimnetic and whole-lake temperature, respectively), and the GCM simulations promoted shorter duration in stratification. The projected increase in spring and autumn hypolimnetic temperature were similar in magnitude and in summer non-significant trends were detected either in this study or in Moras et al. (2019). The simulations made here and by Moras et al (2019) are for the same lake using the same lake model.  The fact that the simulations presented here using the RCP 6.0 emission scenarios showed similar or slightly lower rates of change compared to the simulations made by Moras et al (2019) when the model was forced with measured historical data, are unexpected given that the RCP 6.0 scenario would project an accelerated rate of climate change compared to the historical period.  This suggests that, at least for Lake Erken, future changes in lake thermal structure based on the ISIMIP2b GCM projections may to some extent underestimate the actual changes that will occur.

The projected changes in thermal stratification can influence many aspects of the lake ecosystem. Increases in thermal stability and duration of stratification can intensify hypolimnetic oxygen depletion (Foley et al., 2012; Schwefel et al., 2016) and hence induce enhanced internal phosphorous loading (North et al., 2014), increase the release of dissolved iron and manganese from sediments (Schultze et al., 2017) and also increase methane emissions (Grasset et al., 2018). Warming lake temperature affects biological rates of metabolism, growth and reproduction (Rall et al., 2012) and can promote cyanobacterial blooms (Paerl and Paul, 2012). When coupled to a reduction in oxygen-rich water, warming water temperature leads to a lower fish populations (O'Reilly et al., 2003; Yankova et al., 2017). Increase in evaporation associated with warming can lead to declines in lake water level (Hanrahan et al., 2010) with implications for water security.

## 4. Conclusion

In this study, which was the first test simulating lake hydrothermal structure following ISIMIP2b protocols, ensemble simulations show that changes in Lake Erken's surface temperature was projected to increase on average by 1.79 $^{0}$C for RCP 2.6 and by 3.08 $^{0}$C for RCP 6.0, and the length of the stratification also was projected to be longer by 13 days for RCP 2.6 and by 22 days for RCP 6.0 by the end of the 21$^{st}$ century. Most changes in the RCP 2.6 scenario occurred during the first half of the century suggesting that the aggressive mitigation methods represented in this scenario would be effective at reducing future changes in lake thermal structure. We also extensively document coinciding changes in water column temperatures, water column stability and thermocline depth both in this paper and the supplementary material. Combined these results suggest important changes in the factors affecting lake biogeochemistry directly through changes in temperature and indirectly by influencing the availability of light and nutrients. By providing an initial test for the simulations that will be carried out by the ISIMIP lake sector this paper sets the stage for more extensive world-wide evaluation of the effects of climate change on lake thermal structure.

This study showed the ability of the GOTM model to simulate accurately Lake Erken water temperature when the model was forced using either daily or hourly temporal resolution inputs. Neural networks were shown to be an effective method to disaggregate different weather variables such as air temperature and short-wave radiation, in order to generate synthetic hourly meteorological data from the daily data that are typically available from GCM models. Model performance was slightly improved when using the synthetic hourly data, and climate change effects were somewhat lower when using such data to drive future climate simulations. However, it is not clear that data disaggregation is needed given the computational costs of developing such data sets and running long-term simulations at an hourly time step. Future climate simulations showed similar trends in the anomaly distributions when the GOTM model was forced with mean daily or synthetic hourly meteorological data, and we also found evidence that the calibration procedure partly compensates for differences in the temporal resolution of the model input.

**Code and data availability**

The source code of the model GOTM is freely available online at https://gotm.net/. The input data, model configuration, output and observed data for calibration are stored in HydroShare at https://doi.org/10.4211/hs.ace98c3bc72b44f1834a58ec8b3af310. The ISIMIP climate scenarios are available online at https://www.isimip.org/. Future projections of simulated water temperature derived from both the original ISIMIP input data and synthetic hourly projections are stored in HydroShare at https://doi.org/10.4211/hs.2b4cfe0f02bf4375bcd0b62e45c61b19. Matlab codes, R codes and all the datasets produced during this study are available upon request from the corresponding author.

**Author contributions**

DCP and AIA designed the study. SM provided meteorological data. AIA performed GRNN models and GOTM simulations and analysed the results. AIA wrote the paper with contribution from DCP.

**Competing interests**

The authors declare that they have no conflict of interest.

**Acknowledgements**

We are grateful to ISIMIP for their roles in producing, coordinating, and making available the ISIMIP climate scenarios, we acknowledge the support of the ISIMIP cross sectoral science team. We acknowledge funding from the EU and FORMAS in the frame of the collaborative international Consortium PROGNOS financed under the ERA-NET WaterWorks2014 Cofunded Call. This ERA-NET is an integral part of the 2015 Joint Activities developed by the Water Challenges for a Changing World Joint Programme Initiative (Water JPI). We also acknowledge funding from the European Union's Horizon 2020 research and innovation programme under the Marie Sklodowska-Curie grant agreement H2020-MSCA-ITN-2016 No 722518 for the Project MANTEL, and the project WATExR which is part of ERA4CS, an ERA-NET initiated by JPI Climate, and funded by

MINECO (ES),FORMAS (SE), BMBF (DE), EPA (IE), RCN (NO), and IFD (DK),with co-funding by the European Union (Grant 690462) and FORMAS grant 2017-01738.

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

**Tables**

**Table 1: Bias-corrected variables in the ISIMIP dataset**

| variable name | abbreviation | units |
|---|---|---|
| precipitation | pr | $kg\ m^{-2}\ s^{-1}$ |
| surface pressure | ps | Pa |
| surface downwelling shortwave radiation | rsds | $W\ m^{-2}$ |
| near-surface wind speed | sfcWind | $m\ s^{-1}$ |
| near-surface air temperature | tas | K |
| daily maximum near-surface air temperature | tasmax | K |
| daily minimum near-surface air temperature | tasmin | K |
| relative humidity | hurs | % |

**Table 2: GOTM lake model parameters and calibrated values**

| model parameter | feasible range | calibrated values | | |
|---|---|---|---|---|
| | | 24 h met | 1h met | synthetic 1h met |
| shf_factor | 0.5–1.5 | 0.69 | 0.81 | 0.77 |
| swr_factor | 0.8–1.2 | 1.15 | 0.90 | 0.91 |
| wind_factor | 0.5–2.0 | 1.55 | 1.37 | 1.51 |
| k_min | $1.4\ 10^{-7}$–$1.0\ 10^{-5}$ | $1.47\ 10^{-6}$ | $1.40\ 10^{-6}$ | $1.29\ 10^{-6}$ |
| g2 | 0.5–3.5 | 1.99 | 2.30 | 1.62 |


**Table 3: GRNN models performance evaluation.**

| | MBE | | RMSE | | NSE | |
|---|---|---|---|---|---|---|
| | training | validation | training | validation | training | validation |
| air temperature ($^oC$) | $-1.70 \ 10^{-4}$ | -0.06 | 0.26 | 0.32 | 1.00 | 0.94 |
| short wave radiation (W $m^{-2}$) | $5.76 \ 10^{-4}$ | -0.04 | 6.35 | 8.20 | 1.00 | 0.87 |
| relative humidity (%) | $-7.94 \ 10^{-4}$ | 0.34 | 0.79 | 1.02 | 1.00 | 0.69 |
| wind speed (m $s^{-1}$) | $-5.67 \ 10^{-3}$ | -0.01 | 1.06 | 1.37 | 0.78 | 0.58 |

**Table 4: GOTM lake model performance evaluation: MBE , RMSE and NSE for full profiles temperature, surface temperature, bottom temperature, volumetrically weighted averaged whole lake temperatures, Schmidt stability, thermocline, and duration, onset and loss of stratification using simulated results from running GOTM driven by daily (24h met), hourly (1h met) and synthetic hourly (synthetic 1h met) meteorological data sets.**

| | calibration | | | | | | | | |
|---|---|---|---|---|---|---|---|---|---|
| | MBE | | | RMSE | | | NSE | | |
| | 24h met | 1h met | synthetic 1h met | 24h met | 1h met | synthetic 1h met | 24h met | 1h met | synthetic 1h met |
| full-profile temp ($^oC$) | -0.08 | -0.02 | -0.02 | 1.04 | 0.94 | 0.88 | 0.93 | 0.94 | 0.95 |
| surface temp ($^oC$) | -0.04 | 0.04 | -0.01 | 0.69 | 0.72 | 0.61 | 0.97 | 0.97 | 0.98 |
| bottom temp ($^oC$) | -0.06 | 0.07 | -0.11 | 1.33 | 1.24 | 1.16 | 0.83 | 0.85 | 0.87 |
| whole lake temp ($^oC$) | -0.07 | -0.03 | -0.01 | 0.57 | 0.52 | 0.49 | 0.98 | 0.98 | 0.98 |
| Schmidt stability (J $m^{-2}$) | 0.53 | 0.59 | 0.76 | 22.09 | 21.69 | 19.64 | 0.85 | 0.85 | 0.88 |
| thermocline depth (m) | 0.58 | 0.84 | 0.43 | 2.77 | 3.07 | 2.84 | 0.32 | 0.22 | 0.32 |
| duration (day) | 0.25 | 3.75 | 6.63 | 9.25 | 14.25 | 14.94 | - | - | - |
| onset (day) | -0.63 | -0.50 | -0.13 | 1.54 | 1.12 | 1.17 | - | - | - |
| loss (day) | -0.63 | -2.00 | -2.88 | 1.54 | 5.87 | 9.13 | - | - | - |
| | validation | | | | | | | | |
| | MBE | | | RMSE | | | NSE | | |
| | 24h met | 1h met | synthetic 1h met | 24h met | 1h met | synthetic 1h met | 24h met | 1h met | synthetic 1h met |
| full-profile temp ($^oC$) | -0.07 | -0.12 | 0.00 | 0.63 | 0.69 | 0.68 | 0.98 | 0.97 | 0.97 |
| surface temp ($^oC$) | -0.24 | -0.19 | -0.15 | 0.54 | 0.64 | 0.54 | 0.99 | 0.98 | 0.99 |
| bottom temp ($^oC$) | 0.16 | 0.09 | 0.23 | 0.68 | 0.59 | 0.74 | 0.96 | 0.97 | 0.95 |
| whole lake temp ($^oC$) | -0.13 | -0.17 | -0.06 | 0.48 | 0.59 | 0.51 | 0.99 | 0.98 | 0.98 |
| Schmidt stability (J $m^{-2}$) | -5.26 | -3.26 | -4.47 | 13.27 | 13.50 | 13.26 | 0.90 | 0.90 | 0.90 |

| | | | | | | | | | |
|---|---|---|---|---|---|---|---|---|---|
| thermocline depth (m) | 0.89 | 1.07 | 0.98 | 2.86 | 3.27 | 3.18 | 0.07 | -0.07 | -0.14 |
| duration (day) | -4.50 | -3.50 | -4.50 | 8.75 | 8.28 | 7.11 | - | - | - |
| onset (day) | 0.50 | -7.50 | 0.50 | 0.71 | 10.61 | 0.71 | - | - | - |
| loss (day) | -4.00 | 13.00 | -4.00 | 8.94 | 15.26 | 7.21 | - | - | - |

**Table 5.Trend analysis from 2011-2100 for surface temperature, bottom temperature, whole-lake temperature, Schmidt stability, thermocline depth, duration, onset and loss of stratification (ns: not significant) for RCP 6.0.**

| | | RCP 6.0 | | | |
|---|---|---|---|---|---|
| | | 24h met | | 1h met | |
| | | rate (decade$^{-1}$) | p-value | rate (decade$^{-1}$) | p-value |
| surface temperature ($^o$C) | GFDL-ESM2M | 0.16 | $< 0.001$ | 0.12 | $< 0.001$ |
| | HadGEM2-ES | 0.38 | $< 0.001$ | 0.27 | $< 0.001$ |
| | IPSL-CM5A-LR | 0.38 | $< 0.001$ | 0.29 | $< 0.001$ |
| | MIROC5 | 0.29 | $< 0.001$ | 0.22 | $< 0.001$ |
| bottom temperature ($^o$C) | GFDL-ESM2M | | ns | | ns |
| | HadGEM2-ES | 0.06 | $< 0.05$ | | ns |
| | IPSL-CM5A-LR | | ns | | ns |
| | MIROC5 | 0.11 | $< 0.001$ | 0.09 | $< 0.01$ |
| whole-lake temperature ($^o$C) | GFDL-ESM2M | 0.14 | $< 0.001$ | 0.10 | $< 0.001$ |
| | HadGEM2-ES | 0.25 | $< 0.001$ | 0.16 | $< 0.001$ |
| | IPSL-CM5A-LR | 0.26 | $< 0.001$ | 0.19 | $< 0.001$ |
| | MIROC5 | 0.23 | $< 0.001$ | 0.18 | $< 0.001$ |
| Schmidt stability (J m$^{-2}$) | GFDL-ESM2M | 2.69 | $< 0.01$ | 1.92 | $< 0.01$ |
| | HadGEM2-ES | 7.97 | $< 0.001$ | 6.50 | $< 0.001$ |
| | IPSL-CM5A-LR | 8.15 | $< 0.001$ | 6.36 | $< 0.001$ |
| | MIROC5 | 4.23 | $< 0.01$ | 2.93 | $< 0.01$ |
| thermocline depth (m) | GFDL-ESM2M | 0.07 | $< 0.05$ | | ns |
| | HadGEM2-ES | 0.13 | $< 0.001$ | 0.13 | $< 0.001$ |
| | IPSL-CM5A-LR | *0.05* | *0.06* | 0.09 | $< 0.01$ |
| | MIROC5 | | ns | | ns |
| duration (days) | GFDL-ESM2M | | ns | | ns |
| | HadGEM2-ES | 3.56 | $< 0.001$ | 3.08 | $< 0.001$ |
| | IPSL-CM5A-LR | 3.16 | $< 0.001$ | 2.50 | $< 0.001$ |

| | | | | | | |
|---|---|---|---|---|---|---|
| | MIROC5 | 2.45 | < 0.001 | 2.00 | < 0.001 | |
| onset (day) | GFDL-ESM2M | | ns | | ns | |
| | HadGEM2-ES | -1.95 | < 0.001 | -1.43 | < 0.001 | |
| | IPSL-CM5A-LR | -1.98 | < 0.001 | -1.48 | < 0.001 | |
| | MIROC5 | -1.80 | < 0.001 | -1.45 | < 0.001 | |
| loss (day) | GFDL-ESM2M | | ns | | ns | |
| | HadGEM2-ES | 1.83 | < 0.001 | 1.42 | < 0.001 | |
| | IPSL-CM5A-LR | 1.31 | < 0.001 | 1.06 | < 0.001 | |
| | MIROC5 | 0.83 | < 0.001 | 0.52 | < 0.01 | |


**Table 6. Average thermal metrics for reference period (1981-2010), and average projected change in thermal metrics for mid-century and late-century for RCP 6.0.**

| | | RCP 6.0 | | | | | |
|---|---|---|---|---|---|---|---|
| | | 24h met | | | 1h met | | |
| | | reference period | mid-century | late century | reference period | mid-century | late century |
| surface temperature ($^{\circ}$C) | GFDL-ESM2M | 13.71 | 1.03 | 1.67 | 13.89 | 0.81 | 1.28 |
| | HadGEM2-ES | 13.56 | 3.04 | 4.04 | 13.84 | 2.29 | 2.98 |
| | IPSL-CM5A-LR | 13.64 | 2.60 | 3.62 | 13.86 | 1.92 | 2.69 |
| | MIROC5 | 13.41 | 1.98 | 2.97 | 13.69 | 1.52 | 2.28 |
| | ensemble | 13.58 | 2.16 | 3.08 | 13.82 | 1.64 | 2.31 |
| bottom temperature ($^{\circ}$C) | GFDL-ESM2M | 9.23 | 0.94 | 1.24 | 9.66 | 0.68 | 0.90 |
| | HadGEM2-ES | 9.32 | 0.42 | 0.91 | 9.75 | 0.15 | 0.42 |
| | IPSL-CM5A-LR | 9.29 | 1.18 | 1.19 | 9.61 | 0.88 | 0.79 |
| | MIROC5 | 8.94 | 0.98 | 1.18 | 9.34 | 0.77 | 0.90 |
| | ensemble | 9.19 | 0.88 | 1.13 | 9.59 | 0.62 | 0.75 |
| whole-lake temperature ($^{\circ}$C) | GFDL-ESM2M | 12.44 | 1.06 | 1.61 | 12.83 | 0.79 | 1.20 |
| | HadGEM2-ES | 12.44 | 1.96 | 2.81 | 12.89 | 1.45 | 1.98 |
| | IPSL-CM5A-LR | 12.36 | 2.12 | 2.75 | 12.72 | 1.57 | 1.99 |
| | MIROC5 | 12.11 | 1.69 | 2.41 | 12.59 | 1.27 | 1.84 |
| | ensemble | 12.34 | 1.71 | 2.39 | 12.76 | 1.27 | 1.75 |
| Schmidt stability (J m$^{-2}$) | GFDL-ESM2M | 69.90 | 4.94 | 12.26 | 65.42 | 4.50 | 9.79 |
| | HadGEM2-ES | 66.43 | 59.78 | 77.57 | 63.18 | 48.50 | 61.43 |
| | IPSL-CM5A-LR | 67.52 | 38.67 | 64.62 | 65.73 | 28.06 | 49.23 |

| | | | | | | | |
|---|---|---|---|---|---|---|---|
| | MIROC5 | 68.96 | 23.08 | 42.42 | 66.39 | 17.49 | 31.83 |
| | ensemble | 68.20 | 31.62 | 49.22 | 65.18 | 24.64 | 38.07 |
| thermocline depth (m) | GFDL-ESM2M | -7.82 | -0.39 | -0.22 | -8.50 | -0.17 | -0.08 |
| | HadGEM2-ES | -8.23 | 1.02 | 1.26 | -8.77 | 0.98 | 1.23 |
| | IPSL-CM5A-LR | -7.83 | 0.28 | 0.59 | -8.26 | 0.24 | 0.64 |
| | MIROC5 | -7.83 | 0.09 | 0.34 | -8.51 | 0.28 | 0.49 |
| | ensemble | -7.93 | 0.25 | 0.49 | -8.51 | 0.33 | 0.57 |
| duration (days) | GFDL-ESM2M | 126 | -9 | -8 | 129 | -10 | -9 |
| | HadGEM2-ES | 123 | -8 | -8 | 126 | -9 | -9 |
| | IPSL-CM5A-LR | 123 | -8 | -8 | 126 | -9 | -8 |
| | MIROC5 | 124 | -8 | -8 | 128 | -9 | -9 |
| | ensemble | 124.08 | -11 | -11 | 127 | -12 | -12 |
| onset (day) | GFDL-ESM2M | 131 | -10 | -10 | 131 | -11 | -11 |
| | HadGEM2-ES | 133 | -10 | -10 | 133 | -11 | -11 |
| | IPSL-CM5A-LR | 133 | -10 | -10 | 133 | -11 | -11 |
| | MIROC5 | 134 | -11 | -10 | 133 | -11 | -11 |
| | ensemble | 133 | -65 | -58 | 133 | -61 | -56 |
| loss (day) | GFDL-ESM2M | 257 | -7 | 11 | 263 | -15 | -2 |
| | HadGEM2-ES | 255 | -29 | -3 | 258 | -38 | -16 |
| | IPSL-CM5A-LR | 260 | -46 | -27 | 263 | -49 | -35 |
| | MIROC5 | 257 | -37 | -19 | 260 | -41 | -27 |
| | ensemble | 257 | 7 | 8 | 261 | 8 | 8 |

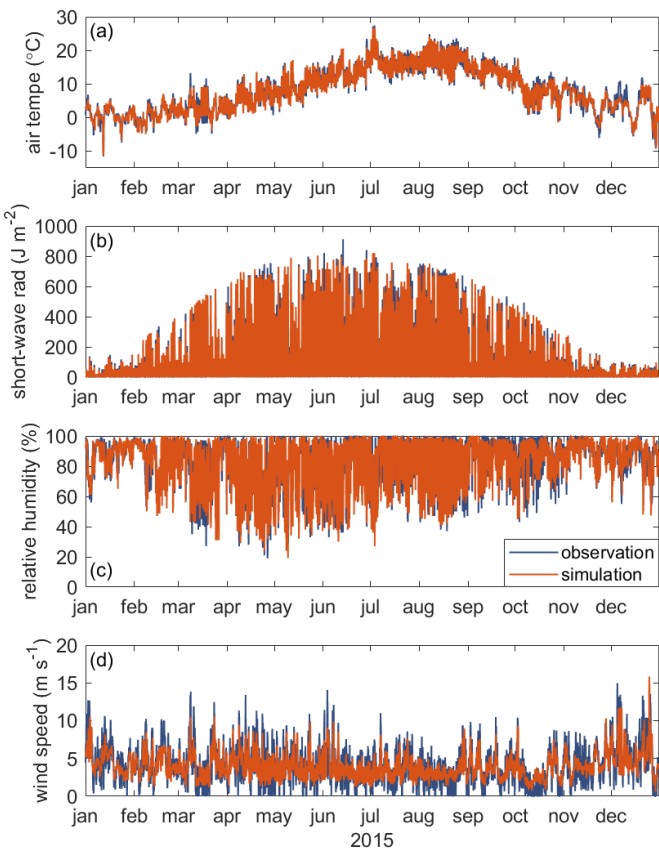

**Figure 1. GRNN temporal disaggregation of meteorological forcing data. Observations vs simulations (a) air temperature, (b) short-wave radiation, (c) relative humidity and (d) wind speed for 2015 (validation data set).**

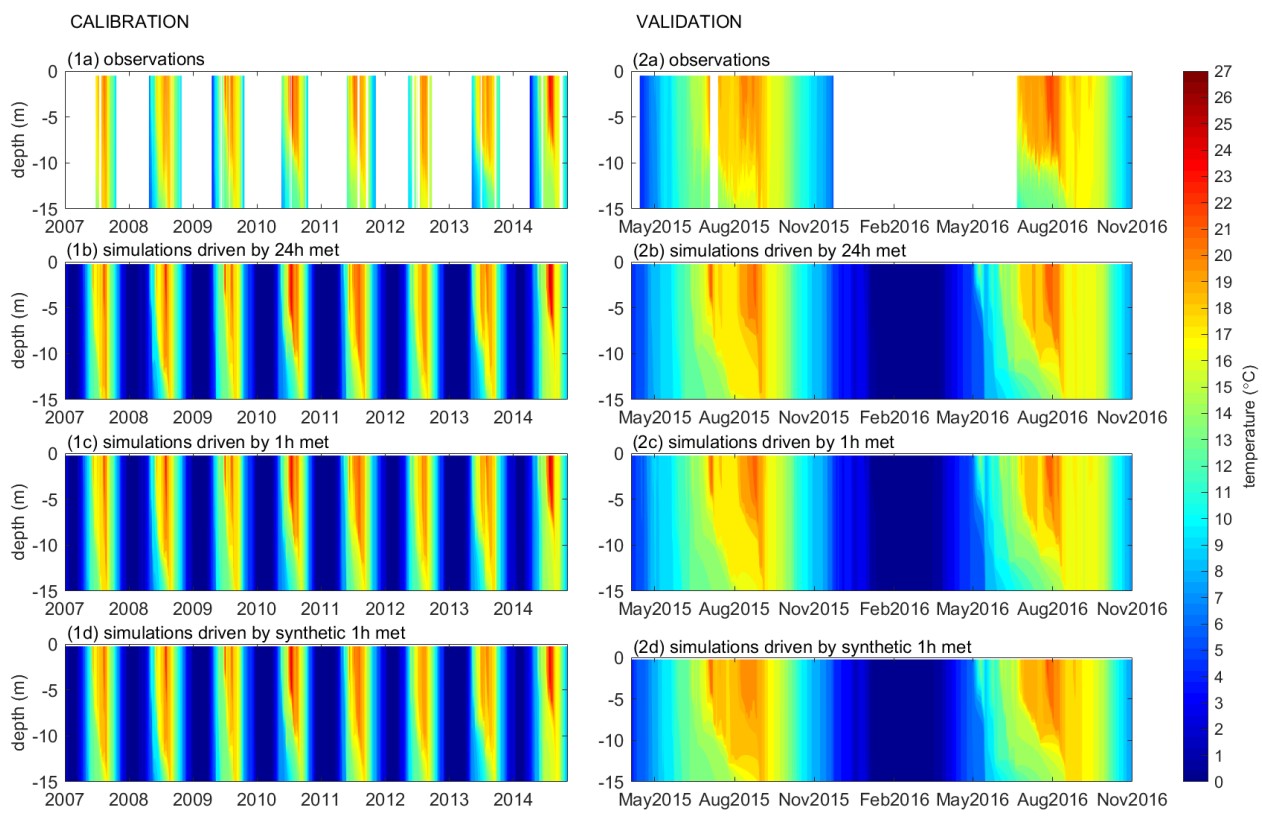


**Figure 2. GOTM water temperature simulations. Daily averaged water temperature in Lake Erken for the calibration (1a)-(1b)-(1c)-(1d) and validation (2a)-(2b)-(2c)-(2d) periods: observations (1a)-(2a), simulations driven by daily meteorological data (1b)-(2b), hourly meteorological data (1c)-(2c) and synthetic hourly meteorological data (1d)-(2d).**

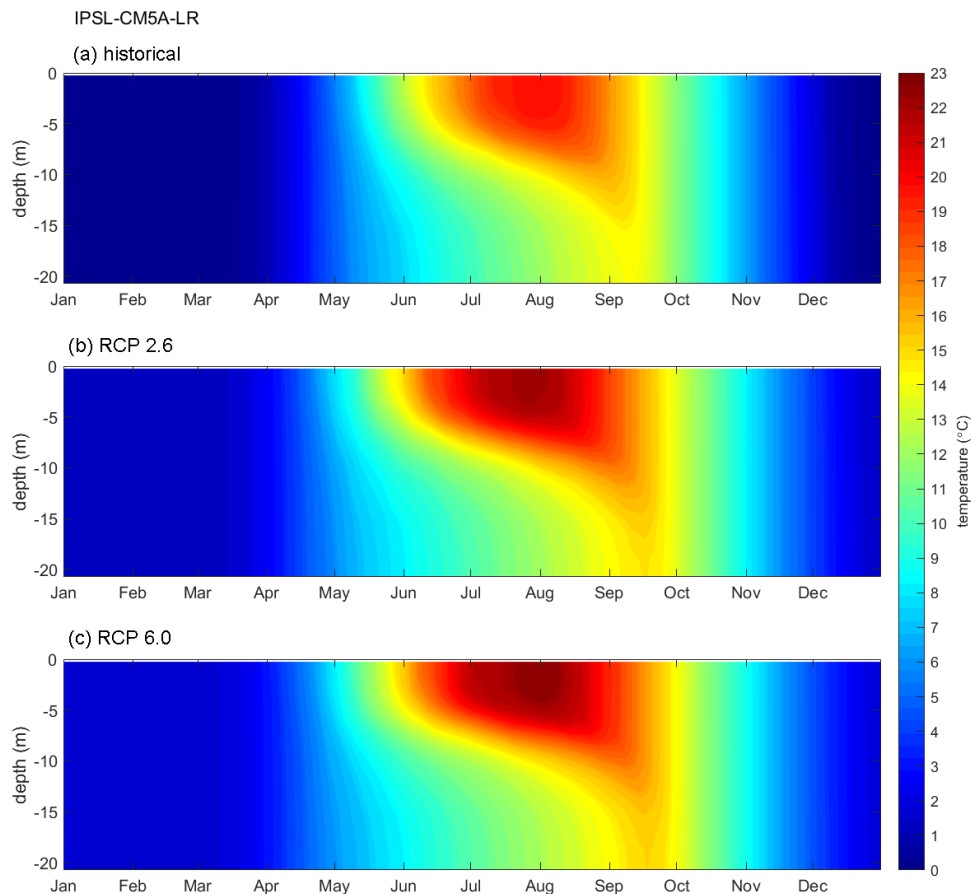

**Figure 3. Temperature isopleth diagrams for the (a) historical, (b) RCP 2.6 and (c) RCP 6.0 scenarios showing results from the lake model forced with daily IPSL-CM5A-LR projections. The temperature matrix used to make these plots was created by averaging the simulated daily temperature profiles for every year in each scenario.**

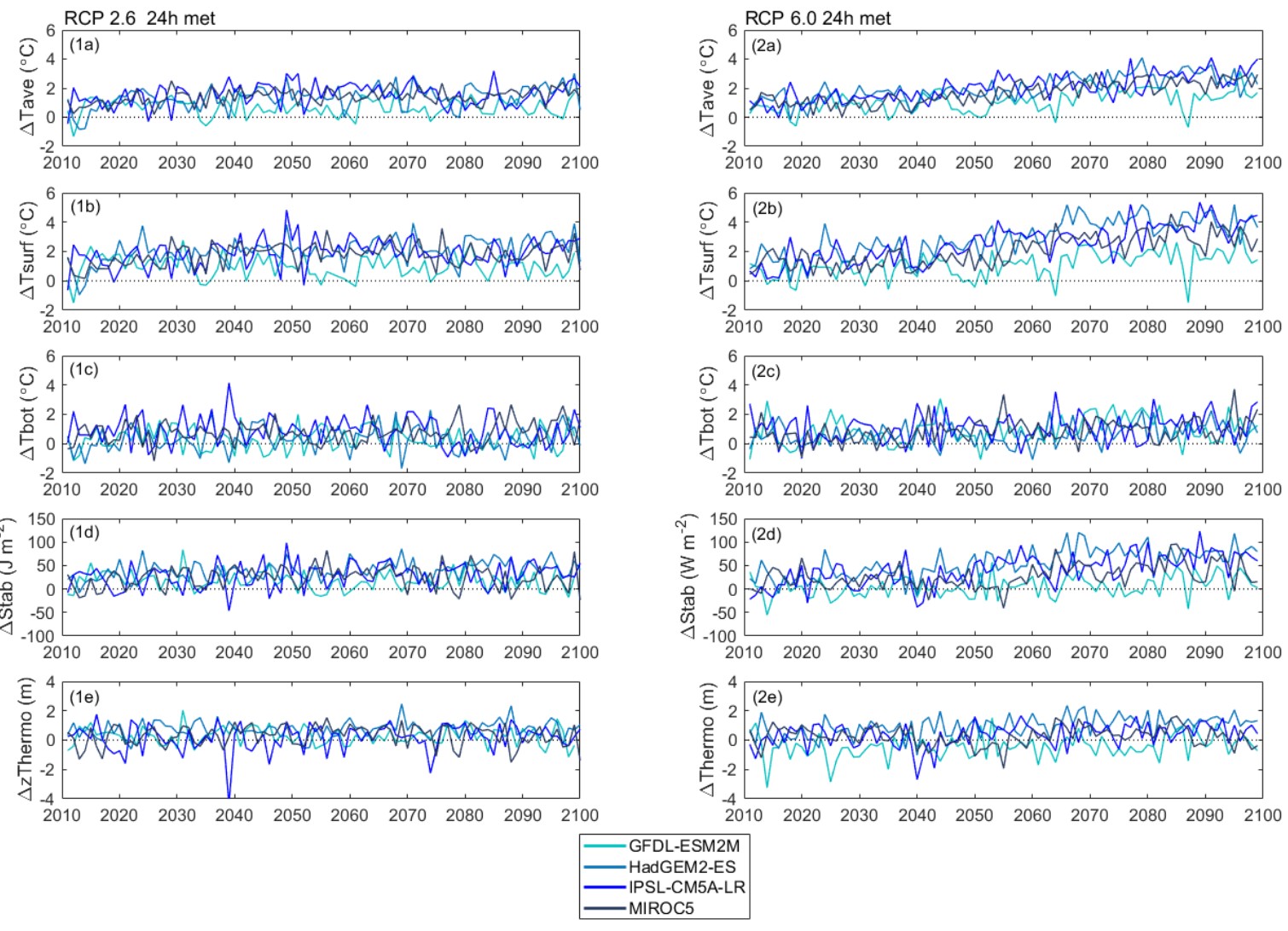

**Figure 4. Evolution of annual average projected anomalies (from April to September) for (1a)-(2a) whole-lake temperature, (1b)-(2b) surface temperature, (1c)-(2c) bottom temperature, (1d)-(2d) Schmidt stability and (1e)-(2e) thermocline depth showing results when the lake model was forced with daily GFDL-ESM2M, HadGEM2-ES, IPSL-CM5A-LR and MIROC5 projections from 2011 to 2100 under RCP 2.6 and 6.0. Anomalies are relative to reference period (1981-2010).**

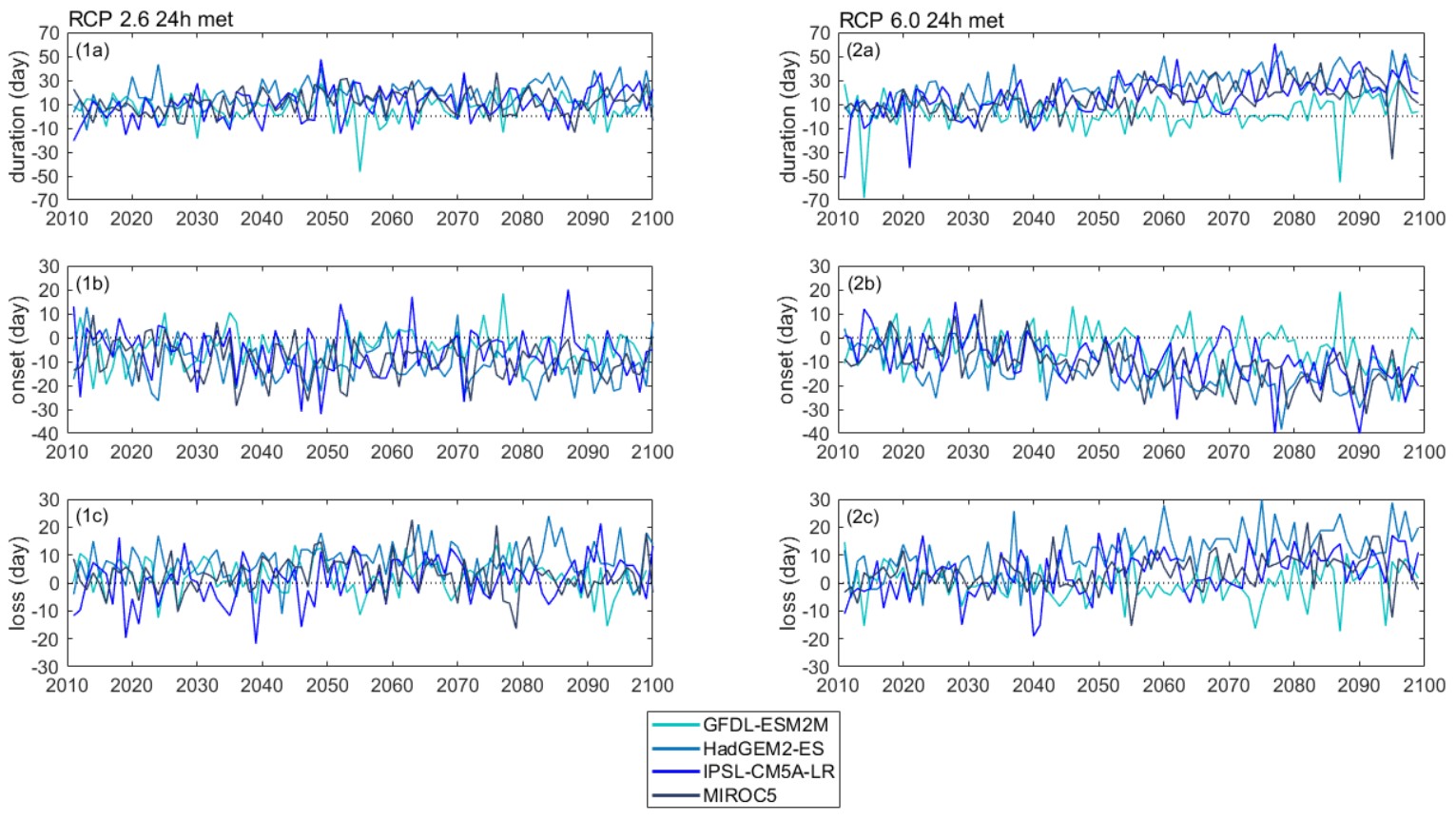

**Figure 5. Evolution of annual average projected anomalies (from April to September) for (1a)-(2a) duration, (1b)-(2b) onset and (1c)-(2c) loss of stratification showing results when the lake model was forced with daily GFDL-ESM2M, HadGEM2-ES, IPSL-CM5A-LR and MIROC5 projections from 2011 to 2100 under RCP 2.6 and 6.0. Anomalies are relative to reference period (1981-2010).**

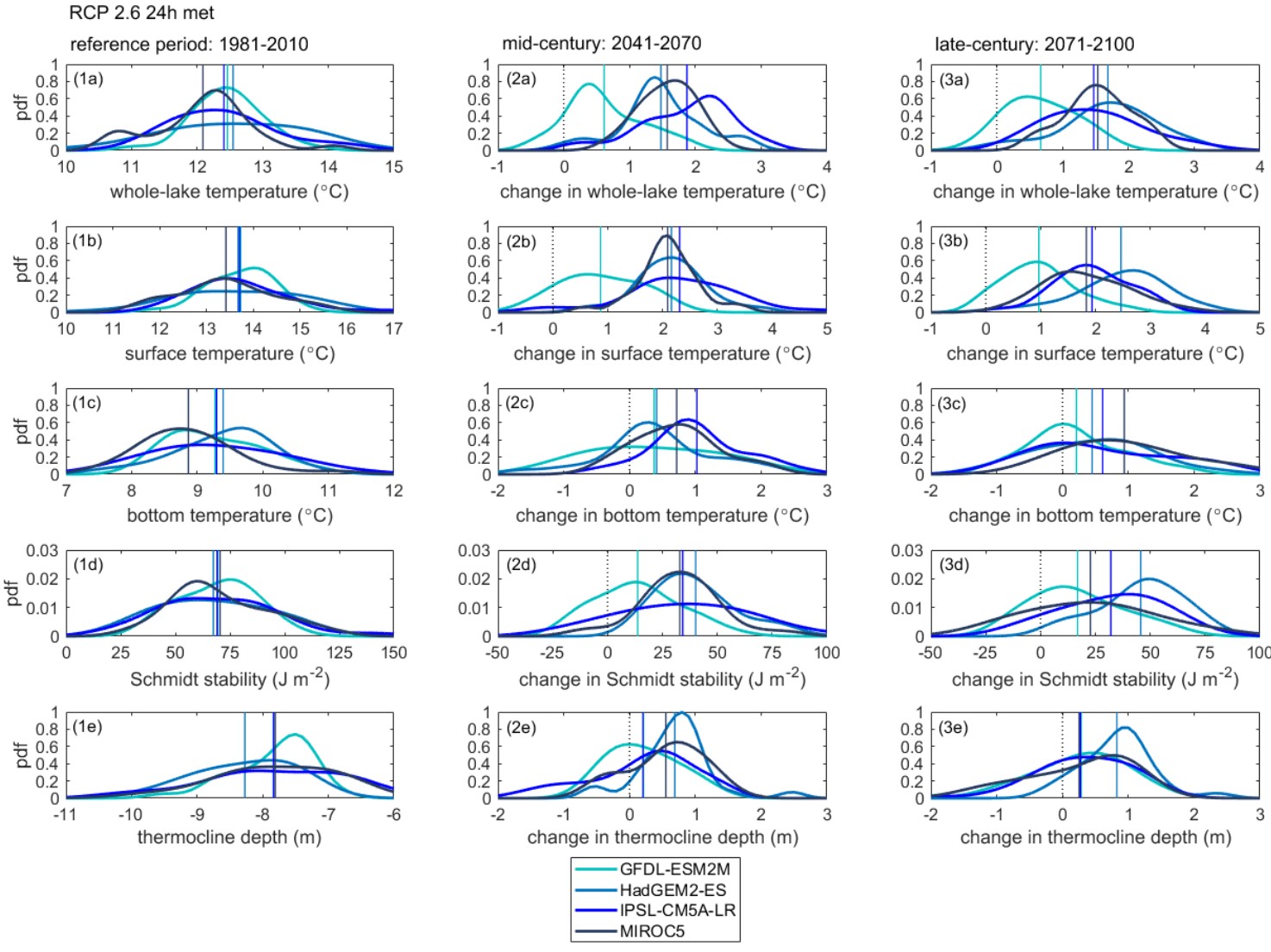

**Figure 6. Changes in annually averaged thermal metrics (from April to September) (2a)-(3a) whole-lake temperature, (2b)-(3b) surface temperature, (2c)-(3c) bottom temperature, (2d)-(3d) Schmidt stability and (2e)-(3e) thermocline depth under RCP 6.0, showing results when the lake model was forced with daily GFDL-ESM2M, HadGEM2-ES, IPSL-CM5A-LR and MIROC5 projections. All changes are for mid-century (2041-2070) and late-century (2071-2100) are relative to reference period (1981-2011). The mean (vertical line) is also shown. Changes in thermal metrics greater than 0 show an increase and lower than 0 show a decrease.**


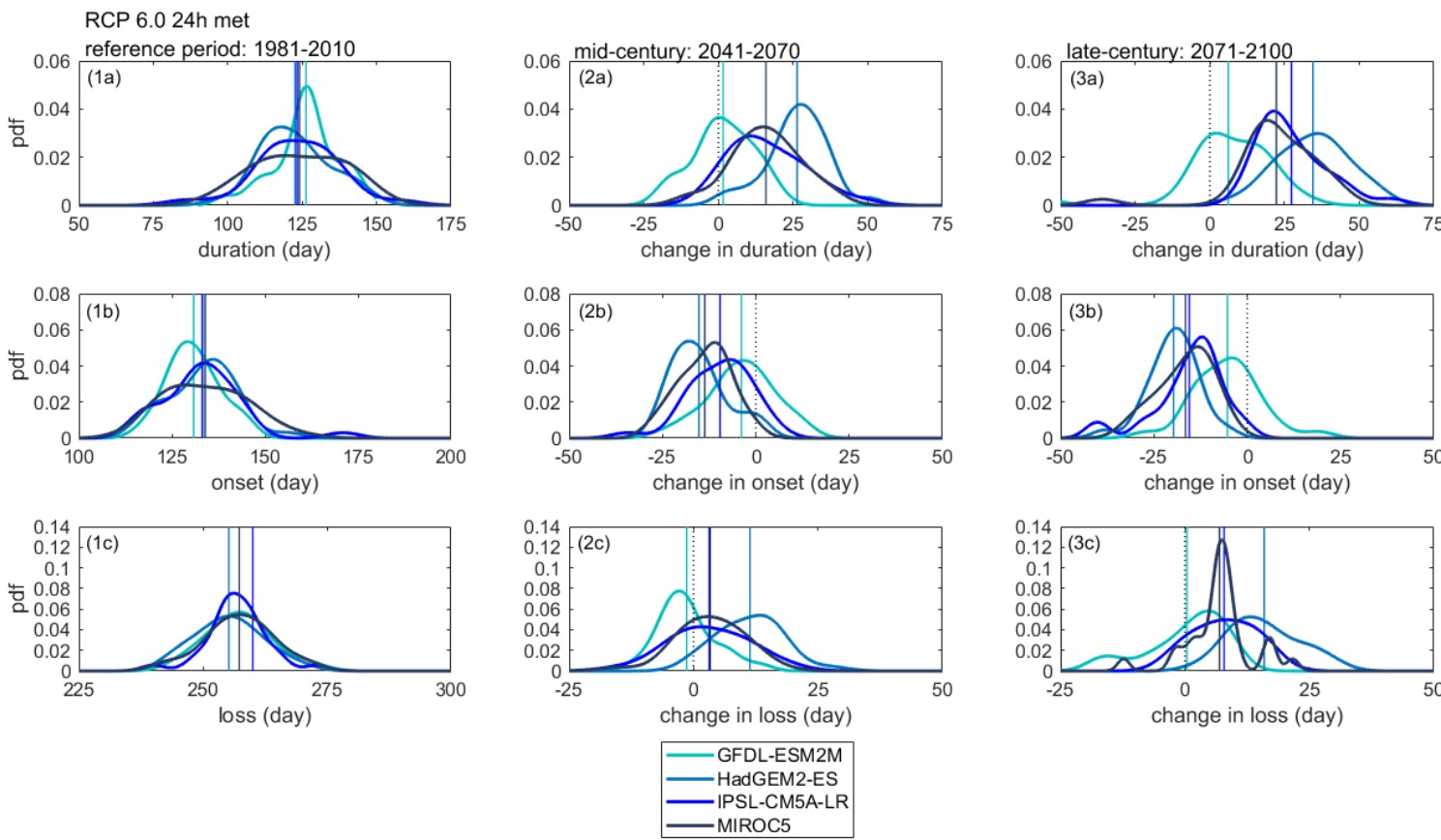

**Figure 7. Changes in annually averaged thermal metrics (from April to September) (2a)-(3a) duration, (2b)-(3b) onset and (2c)-(3c) loss of stratification under RCP 6.0, showing results when the lake model was forced with daily GFDL-ESM2M, HadGEM2-ES, IPSL-CM5A-LR and MIROC5 projections. All changes are for mid-century (2041-2070) and late-century (2071-2100) are relative to reference period (1981-2011). The mean (vertical line) is also shown. Changes in thermal metrics greater than 0 show and increase and lower than 0 show a decrease.**
