# Peer review of "Simulations of future changes in thermal structure of Lake Erken: Proof of concept for ISIMIP2b lake sector local simulation strategy"

_Hydrology and Earth System Sciences, 2019_

## Referee Comment (RC1) · Anonymous Referee #1 · 23 Jul 2019

General comments

The manuscript entitled "Simulations of future changes in thermal structure of Lake Erken: Proof of concept for ISIMIP2b lake sector local simulation strategy" aims to assess the impacts global warming on the thermal characteristics of Lake Erken. Since future projections of global warming are at a daily time step, the authors first analyse the need to disaggregate the input variables to the hourly time step. The manuscript is well written. The topic is scientifically relevant.

Specific comments

Line 70-71: "It is the lake's relatively shallow depth and large surface area, which leads

to large inter-annual variability in the timing and patterns of thermal stratification." Why is this? Perhaps explain in one or two sentences how this works and why this is different for relatively deep lakes or lakes with a small surface area.

Line 138-139: "More detailed description of the GRNN methods and models are given in the supplementary material to this paper." I was hoping to find equations on how the GRNN model calculates hourly estimations based on daily input, however, I could not find a detailed description of the GRNN methods in the supplementary materials.

Line 158: "Schmidt stability", perhaps give a definition or equation of the Schmidt stability

Line 170-174: "Air temperature, short-wave radiation, relative humidity and wind speed were temporarily disaggregated into hourly values from mean daily data, using the GRNN models. A database was constructed using 8 years of measurements. From this whole set of data, the first 5-years of data, that is, from 2008 to 2012, were used for training, and 3-years of data from 2013 to 2015 were used for validating the results obtained." This sentence was confusing. After reading the methods section I first assumed this was about the calibration/validation of GOTM. However, later I realized it was about the calibration/validation of GRNN. I would expect these sentences in the methods section. Moreover, it would be good to mention clearly that there are two types of calibration/validation: that of GOTM and that of GRNN.

Line 192-193: "Temperature simulations for the validation period were more accurate (average RMSE of 0.66 ‰C and NSE of 0.97) than for the calibration period (average RMSE of 0.95 ‰C and NSE of 0.94), but in both periods the model performance was considered acceptable." I would expect that the validation period would be less accurate than the calibration period. Therefore, my first thought was then that perhaps the legend was swapped between calibration and validation. Yet, the authors later mention that this is "due the higher variability in observed water temperature during the long calibration period." (Line 284-285). Then the question raises, which data set

is more representative? Was the high variability during the calibration period actually quite normal and the validation period exceptionally uniform? And what does this mean for the validity of the output?

Line 202: "As would be expected the simulations of bottom temperature were slightly less accurate" Why would this be expected?

Line 349-351: "Combined these results suggest important changes in the factors affecting lake biogeochemistry directly through changes in temperature and indirectly by influencing the availability of light and nutrients." The presented results only indicate an increase in temperature and stratification period. Since the presented data does not show how this affects biogeochemistry and the availability of light and nutrients, could the authors be a bit more specific on this in the conclusion? How do the authors think/speculate it will change (perhaps refer to the introduction where a short explanation is already given)?

Technical corrections

Figures in general; 1) it would be good to have comparable axes per figure. For example, figure 3a has a y-axis going from 0-1.2 oC, while figure 3b goes from 0-0.8 oC. I suggest that the authors uniform the axis and perhaps use the normalized RMSE to compare the different subfigures 2) From the figures caption, it is not always clear if the predicted output is with GOTM or with GRNN. Perhaps include this information in the figure's caption. General: sometimes I read "wind_factor" and sometimes "wind factor" without "_". Is there a difference in meaning?

Line 246-247: "Simulated changes were generally slight less for the simulations driven by daily forcing data as shown by the figures in parentheses" . Put a dot after parentheses and change "slight" to "slightly"

Line 284: "were more accurate than for the calibration period (2006-2014) due the higher variability in observed water temperature" add "to" after "due"

[Figure]

Figure 2: 1) the caption says that validation is figure 2a, 2c, 2e and 2g, however, the title of the figures suggest that validation is figure 2b, 2d, 2f and 2h. This is confusing. 2) Perhaps include the words "observations", "daily data", "hourly data", "synthetic hourly data" on the left side of/ or under the figure. It is now quite a puzzle to find which subfigure tells what. 3) Perhaps also include a difference graph where the difference between "observations" and respectively "daily data", "hourly data", "synthetic hourly data" is shown. From figure 2, it is now hard to see the differences. (The same holds for figure 4, where it is hard to see the differences between historical and the rcp's)

Figure 5 and 6: In figure 5i, the authors indicate the words "deeper" and "shallower" with arrows. This really increases the readability of that specific subfigure and the same would help the reader in all other subfigures.

---

## Referee Comment (RC2) · Anonymous Referee #2 · 21 Oct 2019

General comments

The article "Simulations of future changes in thermal structure of Lake Erken: Proof of concept for ISIMIP2b lake sector local simulation strategy" presents impacts of changing climate on lake water temperature. The article is of very high interest and very well written, the work is thoroughly executed and discussion is relevant. The authors used a hydrodynamic lake model GOTM with 4 GCM/RCMs, using Generalized Regression Artificial Neural Network to disaggregate daily climate into hourly data. The GOTM model was able to reproduce observed lake temperature data for current time period (8 years). The model was then executed with climate forcing data from 4 GCM/RCMs

form ISIMIP2b. The climate impact was evaluated for a set of thermal indices.

Specific comments

I would recommend expanding on the Methods section to provide more information that is critical in understanding the study, its aims, and results. It is unclear why the authors chose to consider 2006-2099 as the future even though the period begins 13 years ago. It is also unclear why this full period is evaluated without any consideration of the changes that occur from 2006 to 2099 based on the trend analyses also included in the manuscript. It seems that changes that can occur during this "future" period are considered representative of changes that will occur by 2099. The averages from this 94-year period are compared to averages from a 30-year period of 1975-2005. The variability during a 30-year period and during a 94-year period with a significant trend is expected to differ and this affects the projected changes.

A more typical approach in many climate impact studies is to select two 30-year periods, one that represents a current climate (reference period, e.g. 1981-2010) and one that represents a future climate (e.g., mid-century 2041-2010 or late century 2071-2100). Forcing data from the same climate model would then be used as model inputs for both time periods; the difference between these results would represent the projected impact. It is also not clear from the manuscript how were the reference period values calculated for calculation of anomalies from the respective GCM/RCMs during the reference time period.

The results for the mid-century and late century should be added to the manuscript to evaluate how the change progresses; alternatively, the current results can be replaced with the late century period as that seem to be the focus of the "proof of concept" study.

It is also important to include information on the variability of the simulated thermal indices due to the climate model selection, i.e. present information for all 4 GCM/RCMs for the reference time period. That can give indication to the significance of the projected impact.
The information on the GOTM model for Lake Erken is very limited; the methods section should be expanded to include more details on the model structure, e.g. vertical resolution, inflow and outflow from the lake, etc.

Some relevant parts should be moved from Results to Methods (e.g., the beginning of section 3.1 and 3.3. Also, the periods used in the calibration (training) & validation periods for GOTM and GRNN should be put into the context between these two models. It is not readily apparent from the manuscript.

It should be emphasized that training the temperature disaggregation algorithm on the current diurnal patterns means those current patterns will be projected to the future time series and any potential changes in diurnal pattern from the changing climate are ignored.

Technical comments

Increases are given to 0.01 deg.C – what is the accuracy of the measurement and of the simulations? Is this accuracy adequate?

L 68 –dimictic?

L89 Mean sea level(,) pressure(,) relative humidity and precipitation were measured – missing commas?

Section 2.6 It would be useful to include model performance for other thermal indices used for evaluation of change, e.g. duration of thermal stratification

L 162: Schmidts stability – needs a reference/ brief explanation

L 231: this model handicap and any other should be described in section 3.2

L 312-314 However, the dominant wind (is) along . . . missing word?

L 322, 324 When GOTM was forcing with . . . forced?

L 350: it would be good to put the statement into context; what kind of changes can be

expected with these increases in temperature?

Figure 2 heading: figure shows calibration as plots a, c, e, and g, but the caption says these are validations

I would recommend including a similar plot but with model residuals (perhaps in Supplementary materials); that would make any differences much easier to see especially on the timing.

Figure 3: it would be helpful if the scale on y axis with the same units had the same range (a-d)

[Figure]

---

## Referee Comment (RC3) · Anonymous Referee #3 · 24 Jan 2020

General comments: The manuscript entitled "Simulations of future changes in thermal structure of Lake Erken: Proof of concept for ISIMIP2b lake sector local simulation strategy" showed the effects of different time-scale forcing data and 4 model forcing and also the 2 RCP future scenario on the simulation with GOTM lake model over Lake Erken. It projected the similar future changing trends of thermal conditions and is helpful for local to understand the effects of climate change and adapt it.

Specific comments:

The work focused on daily characteristics of future thermal contracture in Figure 4-6. The simulated future changing trends are mostly similar with hourly or daily forcing.

But lots of work were done to compare the simulation results with different historical data which may be simplified or removed. Then the work could pay more attention to the future changing characteristics.

L244-246 "Rates of change in whole-lake temperature calculated for over the length for RCP2.6 and 6.0 scenarios were projected to 245 increase except in the case of GFDL-ESM2M which showed weaker or non-significant changes for all measures of thermal stratification." did not match with Table 5.

Some parts were hardly understood, such as "For RCP 6.0, the projected rate of change ranged from 0.15 to 0.27 ‰C decade-1 (0.11 to 0.19 ‰C decade-1). IPSL-CM5A-LR projected the largest increase being 0.59 ‰C (0.43 ‰C) under RCP 2.6 ‰C and 2.51 ‰C (1.79 ‰C) under RCP 6.0.". And IPSL-CM5A-LR did not project the largest temperature increase under RCP 2.6 as showed in Table 5.

Because the lake model parameters are different for different forcing in Table 2. It's hard to know the source of the simulation difference in Table 4 and to evaluate the effects of the time-scale of forcing.

L230 "From these average yearly values were calculated using the months between April and September, due to the fact that the GOTM model was not able to simulate lake ice and winter water temperatures at the same level of accuracy as during the remainder of the year". Does the inaccurate simulation of lake temperature in winter affect the temperature simulation without ice? L68 "The lake is dimictive with summer stratification usually occurring beginning in May-June and ending in August-September, while ice cover occurs from December-February to April-May." Why the average yearly values were calculated including April?

The manuscript was submitted in 2019. It's confused to compare 2006-2099 with 1975-2005 to get the future change.

Does the lake model need downward longwave radiation drive? What's the usage of

[Figure]

**HESSD**

the cloud cover when there is the downward shortwave radiation?

Usually the simulation in the calibration period is better. Why temperature simulations in the validation period were more accurate in the manuscript?

L 110 "under four emission scenarios" As shown in the manuscript, there were only 2 emission scenarios.

If the years for calibration and validation match the years for training and validating, it may be better.

---

## Author Comment (AC1) · 13 Apr 2020

*General comments*
The manuscript entitled "Simulations of future changes in thermal structure of Lake Erken: Proof of concept for ISIMIP2b lake sector local simulation strategy" aims to assess the impacts global warming on the thermal characteristics of Lake Erken. Since future projections of global warming are at a daily time step, the authors first analyses the need to disaggregate the input variables to the hourly time step. The manuscript is well written. The topic is scientifically relevant.
**Response:** We thank the Referee 1 for the positive comments about the text. The paper was edited very carefully and modifications and improvements were made. Below, we address every comment and explain the corresponding changes in the manuscript.

*Specific comments*
Line 70-71: "It is the lake's relatively shallow depth and large surface area, which leads to large inter-annual variability in the timing and patterns of thermal stratification." Why is this? Perhaps explain in one or two sentences how this works and why this is different for relatively deep lakes or lakes with a small surface area.
**Response:** Mixing and stratification change in response to lake morphometry. Shallow lakes have lower heat storage, responding more directly to short-term variations in the weather conditions and heat can be transferred through the water column by wind mixing (Magee and Wu, 2017). However, deep lakes required greater wind speeds to complete the mix. Large surface areas or fetch increase the effects of mixing and vertical transfer of heat to the bottom (Rueda and Schladow, 2009).
**Changes in manuscript:** P3 L74-76.

Line 138-139: "More detailed description of the GRNN methods and models are given in the supplementary material to this paper." I was hoping to find equations on how the GRNN model calculates hourly estimations based on daily input, however, I could not find a detailed description of the GRNN methods in the supplementary materials.
**Response:** GRNN description was added in the supplement material section S1.1.
**Changes in supplement:** P1 L9-34.

Line 158: "Schmidt stability", perhaps give a definition or equation of the Schmidt stability.
**Response:** The following Schmidt stability definition was added: resistance to mechanical mixing due to the potential energy inherent in the density stratification of the water column (Schmidt, 1928; Idso, 1973).
**Changes in manuscript:** P6 L190-191.

Line 170-174: "Air temperature, short-wave radiation, relative humidity and wind speed were temporarily disaggregated into hourly values from mean daily data, using the GRNN models. A database was constructed using 8 years of measurements. From this whole set of data, the first 5-years of data, that is, from 2008 to 2012, were used for training, and 3-years of data from 2013 to 2015 were used for validating the results obtained." This sentence was confusing. After reading the methods section I first assumed this was about the calibration/validation of GOTM. However, later I realized it was about the calibration/validation of GRNN. I would expect these sentences in the methods section. Moreover, it would be good to mention clearly that there are two types of calibration/validation: that of GOTM and that of GRNN.
**Response*:* It has been moved from Results section 3.1. Hourly meteorological modelling to Material and Methods section 2.5. Temporal disaggregation of meteorological forcing data.

**Changes in manuscript:** P5 L147-154.

Line 192-193: "Temperature simulations for the validation period were more accurate (average RMSE of 0.66 °C and NSE of 0.97) than for the calibration period (average RMSE of 0.95 °C and NSE of 0.94), but in both periods the model performance was considered acceptable." I would expect that the validation period would be less accurate than the calibration period. Therefore, my first thought was then that perhaps the legend was swapped between calibration and validation. Yet, the authors later mention that this is "due the higher variability in observed water temperature during the long calibration period." (Line 284-285). Then the question raises, which data set is more representative? Was the high variability during the calibration period actually quite normal and the validation period exceptionally uniform? And what does this mean for the validity of the output?

**Response:** Water temperature simulations were apparently more accurate for the validation period (2015-2016) than for the calibration period (2006-2014), which may appear unusual, but is due to the higher variability in observed water temperature during the longer calibration period. Years with a longer duration of stratification and stronger stability, generally had higher simulation errors. Half of the eight-year calibration period exhibited these conditions, while the two-years used for validation both exhibited shorter duration of stratification and weaker stability.

| | year | RMSE (°C) | | | thermal stratification | | | Schmidt stability (J m$^{-2}$) |
| --- | --- | --- | --- | --- | --- | --- | --- | --- |
| | | 24h met | 1h met | synthetic 1h met | duration (days) | onset | loss | |
| Calibration | 2007 | 0.58 | 0.59 | 0.83 | 23 | 176 | 230 | 17.42 |
| | **2008** | **1.42** | **1.13** | **1.04** | **103** | **124** | **227** | **31.52** |
| | 2009 | 0.75 | 0.68 | 0.63 | 69 | 122 | 242 | 35.17 |
| | **2010** | **1.10** | **0.92** | **0.99** | **111** | **139** | **254** | **80.77** |
| | **2011** | **0.92** | **0.79** | **0.81** | **90** | **152** | **252** | **43.77** |
| | 2012 | 0.71 | 0.66 | 0.77 | 38 | 141 | 244 | 32.98 |
| | **2013** | **1.42** | **1.52** | **1.08** | **124** | **129** | **259** | **79.48** |
| | 2014 | 0.83 | 0.73 | 0.79 | 55 | 137 | 263 | 52.40 |
| Validation | 2015 | 0.59 | 0.66 | 0.65 | 71 | 162 | 240 | 17.60 |
| | 2016 | 0.69 | 0.73 | 0.71 | 67 | 173 | 239 | 47.25 |

**Changes in manuscript:** P14 L433-438.

Line 202: "As would be expected the simulations of bottom temperature were slightly less accurate" Why would this be expected?

**Response:** Higher errors were found at the lowest depth (15 m depth), part of this might have been caused by the presence of internal seiches in lake Erken which cannot be reproduced by 1D models such as GOTM."

Line 349-351: "Combined these results suggest important changes in the factors affecting lake biogeochemistry directly through changes in temperature and indirectly by influencing the availability of light and nutrients." The presented results only indicate an increase in temperature and stratification period. Since the presented data does not show how this affects biogeochemistry and the availability of light and nutrients, could the authors be a bit more specific on this in the conclusion? How do the authors think/speculate it will change (perhaps refer to the introduction where a short explanation is already given)?

**Response:** As mentioned in the introduction the projected changes in thermal stratification can influence many aspects of the lake ecosystem. Increases in thermal stability and duration of stratification can intensify hypolimnetic oxygen depletion (Foley et al., 2012; Schwefel et al., 2016) and hence induce enhanced internal phosphorous loading (North et al., 2014), increase the release of dissolved iron and manganese from sediments (Schultze et al., 2017) and also increase methane emissions (Grasset et al., 2018). Warming lake temperature affects biological rates of metabolism, growth and reproduction (Rall et al., 2012) and can promote cyanobacterial blooms (Paerl and Paul, 2012). When coupled to a reduction in oxygen-rich water, warming water temperature leads to a lower fish populations (O'Reilly et al., 2003; Yankova et al., 2017). Increase in evaporation associated with warming can lead to declines in lake water level (Hanrahan et al., 2010) with implications for water security. So these changes are expected in Lake Erken.

The expected changes in the lake ecosystem caused by changes in thermal stratification have been moved from 1. Introduction to 4. Discussion section and so our conclusions can be more understandable.
**Changes in manuscript:** P18-19 L581-588.

*Technical corrections*
Figures in general; 1) it would be good to have comparable axes per figure. For example, figure 3a has a y-axis going from 0-1.2 °C, while figure 3b goes from 0-0.8 °C. I suggest that the authors uniform the axis and perhaps use the normalized RMSE to compare the different subfigures 2).From the figures caption, it is not always clear if the predicted output is with GOTM or with GRNN. Perhaps include this information in the figure's caption. General: sometimes I read "wind_factor" and sometimes "wind factor" without "_". Is there a difference in meaning?
**Response:**
Figure 3 has been removed because GOTM model performance had been shown in twice (Figure 3 and Table 4).
GRNN and GOTM has been added to the figure captions to indicate if models have been used to disaggregate meteorological forcing data or to simulated water temperature.
Wind factor is the meaning of the parameter wind_factor.

Line 246-247: "Simulated changes were generally slight less for the simulations driven by daily forcing data as shown by the figures in parentheses". Put a dot after parentheses and change "slight" to "slightly"
**Response:** Change made.

Line 284: "were more accurate than for the calibration period (2006-2014) due the higher variability in observed water temperature" add "to" after "due".
**Response:** change made.

Figure 2: 1) the caption says that validation is figure 2a, 2c, 2e and 2g, however, the title of the figures suggest that validation is figure 2b, 2d, 2f and 2h. This is confusing. 2) Perhaps include the words "observations", "daily data", "hourly data", "synthetic hourly data" on the left side of/ or under the figure. It is now quite a puzzle to find which subfigure tells what. 3) Perhaps also include a difference graph where the difference between "observations" and respectively "daily data", "hourly data", "synthetic hourly data" is shown. From figure 2, it is now hard to see the differences. (The same holds for figure 4, where it is hard to see the differences between historical and the rcp's)
**Response:**
Figure 2 has been renumbered and a subtitle added to each subfigure.

Figure S7 has been added to the supplementary material showing the differences between simulated (when the lake model was forced with daily, hourly and synthetic hourly meteorological forcing data) and observed water temperature.

Figure S14 has been added to the supplementary material showing the differences between the historical and RCP 2.6 scenarios, and the historical and RCP 6.0 scenarios for the IPSL-CM5A-LR projection (when the lake model was forced at daily resolutions).

Figure 5 and 6: In figure 5i, the authors indicate the words "deeper" and "shallower" with arrows. This really increases the readability of that specific subfigure and the same would help the reader in all other subfigures.

**Response**: In Figures 6 and 7, the arrow and the words "deeper" and "shallower" have been removed for easy viewing of the figure. However, in the figure caption was added the meaning of values greater or less than 0 of each of the thermal indices: changes in thermal metrics greater than 0 show an increase and lower than 0 show a decrease.

**References:**

Idso, S. B.: On the concept of lake stability, Limnol. Oceanogr., 18, 681–683, 1973.

Magee, M. R., and Wu, C. H.: Response of water temperatures and stratification to changing climate in three lakes with different morphometry, Hydrol. Earth Syst. Sci., 21, 6253-6274, https://doi.org/10.5194/hess-21-6253-2017, 2017.

Rueda, F., and Schladow, G.: Mixing and stratification in lakes of varying horizontal length scales: Scaling arguments and energy partitioning, Limnol. Oceanogr., 54, 2003-2017, https://doi.org/10.4319/lo.2009.54.6.2003, 2009.

Schmidt, W.: Über Temperatur und Stabilitätsverhaltnisse von Seen, Geogr. Ann., 10, 145–177, 1928.

---

## Author Comment (AC2) · 13 Apr 2020

*General comments*

The manuscript entitled "Simulations of future changes in thermal structure of Lake Erken: Proof of concept for ISIMIP2b lake sector local simulation strategy" showed the effects of different time-scale forcing data and 4 model forcing and also the 2 RCP future scenario on the simulation with GOTM lake model over Lake Erken. It projected the similar future changing trends of thermal conditions and is helpful for local to understand the effects of climate change and adapt it.

**Response:** We thank the Referee 3 for the positive comments about the text. The paper was edited very carefully and modifications and improvements were made. Below, we address every comment and explain the corresponding changes in the manuscript.

*Specific comments*

The work focused on daily characteristics of future thermal contracture in Figure 4-6. The simulated future changing trends are mostly similar with hourly or daily forcing. But lots of work were done to compare the simulation results with different historical data which may be simplified or removed. Then the work could pay more attention to the future changing characteristics.

**Response:** The purpose of this paper is twofold: (1) evaluate the importance of diurnal forcing in 1D lake model and (2) assess the long-term impacts of climate change on the thermal structure of Lake Erken. Therefore, we do not consider it appropriate to simplify or remove the first purpose. The difference in mean lake conditions between the reference periods and both mid-century and late-century and long-term trend analysis has been analysed for the climate data and thermal metrics. And also the differences of each meteorological variable and thermal metric were evaluated when the lake model was forced at daily and hourly resolutions respectively.

**Changes in manuscript:** Material and methods: section 2.8 Statistical analysis P9 L255-279, Results: section 3.3 Climate data projections P11 L324-343, section 3.4 Long-term modelled changes in thermal stratification P11-14 L344-420 and section 3.5 Comparison between long-term thermal metrics derived from daily and hourly climate data P13 L421-428. Discussion: P16-18 L500-580.

**Changes in supplement:** Sections S3-S5.

L244-246 "Rates of change in whole-lake temperature calculated for over the length for RCP2.6 and 6.0 scenarios were projected to 245 increase except in the case of GFDL-ESM2M which showed weaker or non-significant changes for all measures of thermal stratification." did not match with Table 5.

**Response:** We do not agree with this comment. Table S8 and Table 5 show the trend analysis under RCP 2.6 and 6.0 respectively for the period 2011-2100. For RCP 2.6 the whole-lake temperature projected under GFDL-ESM2M shows a non-significant increase, and for RCP 6.0 the project increase associated with GFDL-ESM2M was the lowest of the GCMs. For RCP 6.0 the increase in whole-lake temperature ranged from 0.26 to 0.14 $^{\circ}$C decade$^{-1}$.

Some parts were hardly understood, such as "For RCP 6.0, the projected rate of change ranged from 0.15 to 0.27 $^{\circ}$C decade-1 (0.11 to 0.19 $^{\circ}$C decade-1). IPSLCM5A- LR projected the largest increase being 0.59 $^{\circ}$C (0.43 $^{\circ}$C) under RCP 2.6 $^{\circ}$C and 2.51 $^{\circ}$C (1.79 $^{\circ}$C) under RCP 6.0". And IPSL-CM5A-LR did not project the largest temperature increase under RCP 2.6 as showed in Table 5.

**Response:** We totally agree, sometimes it's hard to understand. The results have been rewritten, reducing the large amount of numbers in the text, making it more readable. All the results can be found in the Figures and Tables of both the manuscript and the supplement material. IPSL-CM5A-LR did not project the largest temperature increase under RCP 2.6, under scenario future RCP 2.6 HadGEM2-ES projected

the largest increase in surface temperature, being 0.15 °C decade$^{-1}$. The trend analysis has been carefully reviewed and the results rewritten.
**Changes in manuscript:** P9-11 L282-323, P14-15 L430-461 and P15-16 L478-499.

Because the lake model parameters are different for different forcing in Table 2. It's hard to know the source of the simulation difference in Table 4 and to evaluate the effects of the time-scale of forcing.
**Response:** One of the purposes of this study was to test the ability of a 1D lake model (GOTM) to simulate daily water temperature using daily vs hourly meteorological data, i.e. evaluate the importance of diurnal forcing in 1D lake model. In all cases the lake model was ran at hourly model computational time step when the meteorological forcing was provided at either daily or hourly frequencies. In each case a separate calibration was run using the same observed data for comparison, simulated output derived from the models forced at daily and hourly resolution. We felt that this was the fairest and most representative way to test how the model would actually be applied with the different forcing data. When GOTM was forced at daily resolutions, there is no diurnal variability in the input, which leads to changes in heat fluxes. However it became apparent that variations in model parameters resulting from the different calibrations compensated for some of the differences between observations and simulations based on the different time-scale of forcing. We now point this out more clearly in the paper.
**Changes in the manuscript:** P17 L530-540.

L230 "From these average yearly values were calculated using the months between April and September, due to the fact that the GOTM model was not able to simulate lake ice and winter water temperatures at the same level of accuracy as during the remainder of the year". Does the inaccurate simulation of lake temperature in winter affect the temperature simulation without ice? L68 "The lake is dimictic with summer stratification usually occurring beginning in May-June and ending in August-September, while ice cover occurs from December-February to April-May." Why the average yearly values were calculated including April?
**Response**: The GOTM model version 5.1 did not have the ability to simulate lake ice, so for this study the inverse stratification period was not analysed. Moras et al., (2019), has shown that despite this limitation, the mode is able to accurately simulate water temperature and the phenology of thermal stratification during the remainder of the year. A new GOTM model version 5.4 with ice-module was released after this project was submitted, allowing to evaluate the effect of the lack of ice module on the onset of the direct stratification. The onset of direct stratification was derived from simulations of water temperature with GOTM version 5.1 and 5.4 from 2006 to 2016. The RMSE between the onset of direct stratification from GOTM version 5.1 and 5.4 was 5.22 days showing a slight impact the lack of ice-module on the onset of the direct stratification.

| onset of direct stratification | |
| --- | --- |
| GOTM v5.1 | GOTM v5.4 |
| 2007-04-27 | 2007-04-16 |
| 2008-04-27 | 2008-04-26 |
| 2009-04-27 | 2009-04-27 |
| 2010-05-01 | 2010-05-13 |
| 2011-04-24 | 2011-04-25 |
| 2012-05-03 | 2012-05-01 |
| 2013-05-09 | 2013-05-09 |
| 2014-04-21 | 2014-04-21 |
| 2015-05-22 | 2015-05-22 |

Annual ice cover observations of the onset and loss of ice cover made at lake Erken since 1941 (Moras et al., 2019) showed a decreased since 1941 by 7.34 day decade[-1] (57 days from 1941 to 2017), consistent with changes in air temperature. For this reason, we consider relevant in our long-term study to include April in our analysis.

The manuscript was submitted in 2019. It's confused to compare 2006-2099 with 1975-2005 to get the future change.
**Response:** we totally agree, the choice of reference period is always controversial because the projected impact depends on it. Initially we used as a reference period the last 30 years of the historical scenario (1975-2005) for each GCM, since from 2006 they were already future projections. However, we have decided to slightly update our reference period to 1981-2010.
The table shows the trend analysis for the period 2006-2100 relative to 1975-2005 and for the period 2011-2100 relative to 1981-2010 for HadGEM2-ES under RCP 6.0. The differences are almost unnoticeable, so we do not consider it necessary to update our reference period to 1990-2019.

| | HadGEM2-ES RCP 6.0 | | | |
|---|---|---|---|---|
| | reference period: 1975-2005 | | reference period: 1981-2010 | |
| | 24h met | 1h met | 24h met | 1h met |
| air temperature (°C) | 0.44 °C dec[-1] | 0.33 °C dec[-1] | 0.43 °C dec[-1] | 0.32 °C dec[-1] |
| surface temperature (°C) | 0.38 °C dec[-1] | 0.28 °C dec[-1] | 0.38 °C de-1 | 0.27 °C dec[-1] |
| bottom temperature (°C) | 0.07 °C dec[-1] | ns | 0.06 °C dec-1 | ns |
| whole-lake temperature (°C) | 0.25 °C dec[-1] | 0.17 °C dec[-1] | 0.25 °C dec[-1] | 0.16 °C dec[-1] |
| Schmidt stability (J m[-2]) | 7.79 J m[-2] dec[-1] | 6.22 J m[-2] dec[-1] | 7.97 J m[-2] dec[-1] | 6.50 J m[-2] dec[-1] |
| thermocline depth (m) | 0.12 m dec[-1] | 0.12 m dec-1 | 0.13 m dec[-1] | 0.13 m dec[-1] |

Does the lake model need downward longwave radiation drive? What's the usage of the cloud cover when there is the downward shortwave radiation?
GOTM internally calculates net long-wave radiation from cloud cover according to Clark et al. (1974). Cloud cover for long-term water temperature simulations was estimated from bias-corrected model data according to Martin and McCutcheon (1999):

$$H_{SW} = H_0 \cdot a_t \cdot (1 - R_s) \cdot C_a$$

where $H_{sw}$ is the short-wave solar radiation (W · m[-2]), $H_0$ is the amount of radiation reaching the earth's outer atmosphere (W · m[-2]), $a_t$ is an atmospheric transmission term, $R_s$ albedo or reflection coefficient, and $C_a$ is the fraction of solar radiation not absorbed by clouds.

$$C_a = 1 - 0.65 \cdot C_l^2$$

where $C_l$ is the fraction of the sky covered by clouds.
Cloud cover would be:

$$C_l = \sqrt{\frac{1 - \dfrac{H_{SW}}{H_0 \cdot a_t \cdot (1 - -R_s)}}{0.65}}$$

Usually the simulation in the calibration period is better. Why temperature simulations in the validation period were more accurate in the manuscript?

**Response:** Water temperature simulations were apparently more accurate for the validation period (2015-2016) than for the calibration period (2006-2014), which may appear unusual, but is due to the higher variability in observed water temperature during the longer calibration period. Years with a longer duration of stratification and stronger stability, generally had higher simulation errors. Half of the eight-year calibration period exhibited these conditions, while the two-years used for validation both exhibited shorter duration of stratification and weaker stability.

| | year | RMSE (°C) | | | thermal stratification | | | Schmidt stability (J m$^{-2}$) |
| --- | --- | --- | --- | --- | --- | --- | --- | --- |
| | | 24h met | 1h met | synthetic 1h met | duration (days) | onset | loss | |
| Calibration | 2007 | 0.58 | 0.59 | 0.83 | 23 | 176 | 230 | 17.42 |
| | **2008** | **1.42** | **1.13** | **1.04** | **103** | **124** | **227** | **31.52** |
| | 2009 | 0.75 | 0.68 | 0.63 | 69 | 122 | 242 | 35.17 |
| | **2010** | **1.10** | **0.92** | **0.99** | **111** | **139** | **254** | **80.77** |
| | **2011** | **0.92** | **0.79** | **0.81** | **90** | **152** | **252** | **43.77** |
| | 2012 | 0.71 | 0.66 | 0.77 | 38 | 141 | 244 | 32.98 |
| | **2013** | **1.42** | **1.52** | **1.08** | **124** | **129** | **259** | **79.48** |
| | 2014 | 0.83 | 0.73 | 0.79 | 55 | 137 | 263 | 52.40 |
| Validation | 2015 | 0.59 | 0.66 | 0.65 | 71 | 162 | 240 | 17.60 |
| | 2016 | 0.69 | 0.73 | 0.71 | 67 | 173 | 239 | 47.25 |

**Changes in manuscript:** P14 L433-438.

L 110 "under four emission scenarios". As shown in the manuscript, there were only 2 emission scenarios.
**Response:** Change made.

If the years for calibration and validation match the years for training and validating, it may be better. The GRNN training and validation periods do not fit into GOTM calibration periods. Putting these periods in context does not produce significant changes in the GRNN models performance (see table below) but it would entail a high computational cost since changing the GRNN models would require all the GCM scenarios to be disaggregated a second time and all GCM scenarios to be run again using these alternative data.

| | Air temperature (°C) | | | Relative humidity (%) | | | Wind speed (m s$^{-1}$) | | | Short-wave rad (W m$^{-2}$) | | |
| --- | --- | --- | --- | --- | --- | --- | --- | --- | --- | --- | --- | --- |
| | BIAS | RMSE | NSE | BIAS | RMSE | NSE | BIAS | RMSE | NSE | BIAS | RMSE | NSE |
| Training: 2006-2014 | 0.00 | 0.32 | 1.00 | 0.00 | 0.96 | 1.00 | -0.01 | 1.15 | 0.74 | 0.00 | 8.39 | 1.00 |
| Validation: 2015-2016 | 0.03 | 0.70 | 0.95 | 0.44 | 2.09 | 0.69 | -0.07 | 2.50 | 0.60 | 0.08 | 18.15 | 0.86 |
| Training: 2008-2012 | 0.00 | 0.26 | 1.00 | 0.00 | 0.79 | 1.00 | -0.01 | 1.06 | 0.78 | 0.00 | 6.35 | 1.00 |
| Validation: 2013-2015 | -0.06 | 0.32 | 0.94 | 0.34 | 1.02 | 0.69 | -0.01 | 1.37 | 0.58 | -0.04 | 8.20 | 0.87 |

**References:**
Martin, J., and McCutcheon, M.: Hydrodynamics and Transport for Water Quality Modeling, Lewis Publishers, US, 1999.

Clark, N. E., Eber, L., Laurs, R. M., Renner, J. A., and Saur, J. F. T.: Heat exchange between ocean and atmosphere in the Eastern North Pacific for 1961-71, NOAA Technical Report NMFS SSRF-682, U. S. Dept. of Commerce, Washington, DC, 72 pp., 1974.

---

## Author Comment (AC3) · 13 Apr 2020

*General comments*

The article "Simulations of future changes in thermal structure of Lake Erken: Proof of concept for ISIMIP2b lake sector local simulation strategy" presents impacts of changing climate on lake water temperature. The article is of very high interest and very well written, the work is thoroughly executed and discussion is relevant. The authors used a hydrodynamic lake model GOTM with 4 GCM/RCMs, using Generalized Regression Artificial Neural Network to disaggregate daily climate into hourly data. The GOTM model was able to reproduce observed lake temperature data for current time period (8 years). The model was then executed with climate forcing data from 4 GCM/RCMs

**Response:** We thank the Referee 2 for the positive comments about the text. The paper was edited very carefully and modifications and improvements were made. Below, we address every comment and explain the corresponding changes in the manuscript.

*Specific comments*

I would recommend expanding on the Methods section to provide more information that is critical in understanding the study, its aims, and results. It is unclear why the authors chose to consider 2006-2099 as the future even though the period begins 13 years ago. It is also unclear why this full period is evaluated without any consideration of the changes that occur from 2006 to 2099 based on the trend analyses also included in the manuscript. It seems that changes that can occur during this "future" period are considered representative of changes that will occur by 2099. The averages from this 94-year period are compared to averages from a 30-year period of 1975-2005. The variability during a 30-year period and during a 94-year period with a significant trend is expected to differ and this affects the projected changes. A more typical approach in many climate impact studies is to select two 30-year periods, one that represents a current climate (reference period, e.g. 1981-2010) and one that represents a future climate (e.g., mid-century 2041-2010 or late century 2071-2100). Forcing data from the same climate model would then be used as model inputs for both time periods; the difference between these results would represent the projected impact. It is also not clear from the manuscript how were the reference period values calculated for calculation of anomalies from the respective GCM/RCMs during the reference time period.

The results for the mid-century and late century should be added to the manuscript to evaluate how the change progresses; alternatively, the current results can be replaced with the late century period as that seem to be the focus of the "proof of concept" study.

It is also important to include information on the variability of the simulated thermal indices due to the climate model selection, i.e. present information for all 4 GCM/RCMs for the reference time period. That can give indication to the significance of the projected impact.

**Response:**

Climate impact studies can be approached in two ways: (1) assessing the difference in mean lake conditions (for example, mean surface temperature) between the reference periods and both mid-century and late-century (Woolway and Merchant, 2019) or (2) long-term trend analysis (O'Reilly et al., 2015; Shatwell et al, 2019; Moras et al., 2019). The use of anomalies or absolute values in trend analysis does not change the value of the slope. The use of anomalies in frequency distribution figures provide an alternative method of comparing the changes simulated by the future climate scenarios. So we consider it appropriate to combine both approaches. We have also added information about the reference period and the same analysis has been made to the meteorological variables in order to understand the variability of the projected thermal metrics derived from GCMs.

**Changes in manuscript:** Material and methods: section 2.8 Statistical analysis P9 L255-279, Results: section 3.3 Climate data projections P11 L324-343, section 3.4 Long-term modelled changes in thermal stratification P11-14 L344-420 and section 3.5 Comparison between long-term thermal metrics derived from daily and hourly climate data P14 L421-428. Discussion: P16-18 L500-580.
**Changes in supplement:** Sections S3-S5.

The information on the GOTM model for Lake Erken is very limited; the methods section should be expanded to include more details on the model structure, e.g. vertical resolution, inflow and outflow from the lake, etc.

**Response:** The GOTM model version 5.1 was used in this study. The meteorological parameters for running the model were air temperature ($^{o}$C), wind speed (m s$^{-1}$), short-wave radiation (W m$^{-2}$), cloud cover (dimensionless, 0-1), relative humidity (%), atmospheric pressure (hPa) and precipitation (mm day$^{-1}$ or mm hour$^{-1}$). Inflows and outflows were not included in this study, and water level was considered fixed in the simulations. This version of GOTM did not have the ability to simulate lake ice, so for this study the inverse stratification period was not analysed. Moras et al., (2019), has shown that despite this limitation, the mode is able to accurately simulate water temperature and the phenology of thermal stratification during the remainder of the year. The initial conditions for water temperature were derived from a measured vertical profile. GOTM was run at hourly model computational time step, and simulated water temperature was saved as daily mean values each 0.5 m (42 layers).
**Changes in manuscript:** P5-6 L155-163.

Some relevant parts should be moved from Results to Methods (e.g., the beginning of section 3.1 and 3.3. Also, the periods used in the calibration (training) & validation periods for GOTM and GRNN should be put into the context between these two models. It is not readily apparent from the manuscript.
**Response:**
The beginning of sections 3.1 and 3.3 have been moved to 2. Materials and Methods section 2.5 Temporal disaggregation of meteorological forcing data 2.8 Statistical analysis respectively.
The GRNN training and validation periods do not fit into GOTM calibration periods. Putting these periods in context does not produce significant changes in the GRNN models performance (see table below) but it would entail a high computational cost since changing the GRNN models would require all the GCM scenarios to be disaggregated a second time and all GCM scenarios to be run again using these alternative data.

| | Air temperature ($^{o}$C) | | | Relative humidity (%) | | | Wind speed (m s$^{-1}$) | | | Short-wave rad (W m$^{-2}$) | | |
|---|---|---|---|---|---|---|---|---|---|---|---|---|
| | BIAS | RMSE | NSE | BIAS | RMSE | NSE | BIAS | RMSE | NSE | BIAS | RMSE | NSE |
| Training: 2006-2014 | 0.00 | 0.32 | 1.00 | 0.00 | 0.96 | 1.00 | -0.01 | 1.15 | 0.74 | 0.00 | 8.39 | 1.00 |
| Validation: 2015-2016 | 0.03 | 0.70 | 0.95 | 0.44 | 2.09 | 0.69 | -0.07 | 2.50 | 0.60 | 0.08 | 18.15 | 0.86 |
| Training: 2008-2012 | 0.00 | 0.26 | 1.00 | 0.00 | 0.79 | 1.00 | -0.01 | 1.06 | 0.78 | 0.00 | 6.35 | 1.00 |
| Validation: 2013-2015 | -0.06 | 0.32 | 0.94 | 0.34 | 1.02 | 0.69 | -0.01 | 1.37 | 0.58 | -0.04 | 8.20 | 0.87 |

**Changes in manuscript:** P5 L147-154 and P9 L255-263.

It should be emphasized that training the temperature disaggregation algorithm on the current diurnal patterns means those current patterns will be projected to the future time series and any potential changes in diurnal pattern from the changing climate are ignored.

**Response:** GRNNs proved to be an effective method to disaggregate daily GCM forcing to an hourly temporal resolution for different weather variables such as air temperature, short-wave radiation, etc. However, GRNNs require a training phase, in which the diurnal patterns to be learned are presented to the network from historical meteorological measurements, and therefore if there are future changes in diurnal patterns, these cannot be reproduced. In addition, there is a high computational cost of disaggregating and storing the long-term daily climate data into an hourly data set.

**Changes in manuscript:** P17 L539-545.

*Technical comments*

Increases are given to 0.01 °C – what is the accuracy of the measurement and of the simulations? Is this accuracy adequate?

**Response**: The accuracy of thermocouple sensor is approximately ± 0.1 °C and can at times be somewhat better than 0.1. The simulated water temperature is given with 7 decimals. So two decimal places in the GOTM model performance are adequate, and match the best expected performance of our monitoring data.

L 68 –dimictic?

**Response:** Change made.

L89 Mean sea level(,) pressure(,) relative humidity and precipitation were measured – missing commas?

**Response:** Change made.

Section 2.6. It would be useful to include model performance for other thermal indices used for evaluation of change, e.g. duration of thermal stratification

Response:

**Response:** The model performance for the duration, onset and loss of stratification has been added to the section 3.2 Lake Model performance (Table 4).

L 162: Schmidt's stability – needs a reference/ brief explanation

**Response:** The following Schmidt stability definition was added: resistance to mechanical mixing due to the potential energy inherent in the density stratification of the water column (Schmidt, 1928; Idso, 1973).

**Changes in manuscript:** P6 L190-191.

L 231: this model handicap and any other should be described in section 3.2

**Response**: The GOTM model version 5.1 did not have the ability to simulate lake ice, so for this study the inverse stratification period was not analysed. Moras et al., (2019), has shown that despite this limitation, the mode is able to accurately simulate water temperature and the phenology of thermal stratification during the remainder of the year. A new GOTM model version 5.4 with ice-module was released after this project was submitted, allowing to evaluate the effect of the lack of ice module on the onset of the direct stratification. The onset of direct stratification was derived from simulations of water temperature with GOTM version 5.1 and 5.4 from 2006 to 2016. The RMSE between the onset of direct stratification from GOTM version 5.1 and 5.4 was 5.22 days showing a slight impact the lack of ice-module on the onset of the direct stratification.

| onset of direct stratification | |
|---|---|
| GOTM v5.1 | GOTM v5.4 |
| 2007-04-27 | 2007-04-16 |
| 2008-04-27 | 2008-04-26 |
| 2009-04-27 | 2009-04-27 |
| 2010-05-01 | 2010-05-13 |
| 2011-04-24 | 2011-04-25 |
| 2012-05-03 | 2012-05-01 |
| 2013-05-09 | 2013-05-09 |
| 2014-04-21 | 2014-04-21 |
| 2015-05-22 | 2015-05-22 |
| 2016-05-04 | 2016-05-03 |

**Changes in manuscript:** P6 L159-162.

L 312-314 However, the dominant wind (is) along : : : missing word?
**Response:** Change made.

L 322, 324 When GOTM was forcing with : : : forced?
**Response:** Change made.

L 350: it would be good to put the statement into context; what kind of changes can be expected with these increases in temperature?
**Response:** The expected changes in the lake ecosystem caused by an increase in water temperature have been moved from 1. Introduction to 4. Discussion section.

Figure 2 heading: figure shows calibration as plots a, c, e, and g, but the caption says these are validations. I would recommend including a similar plot but with model residuals (perhaps in Supplementary materials); that would make any differences much easier to see especially on the timing.
**Response:**
Figure 2 has been renumbered and a subtitle added to each subfigure.
Figure S7 has been added to the supplementary material showing the differences between simulated (when the lake model was forced with daily, hourly and synthetic hourly meteorological forcing data) and observed water temperature.

Figure 3: it would be helpful if the scale on y axis with the same units had the same range (a-d)
Figure 3 has been removed because GOTM model performance had been shown in twice (Figure 3 and Table 4).

**References:**
Idso, S. B.: On the concept of lake stability, Limnol. Oceanogr., 18, 681–683, 1973.
Moras, S., Ayala, A. I., and Pierson, D. C.: Historical modelling of changes in Lake Erken thermal conditions, Hydrol. Earth Syst. Sci., 23, 5001–5016, https://doi.org/10.5194/hess-23-5001-2019, 2019.
O'Reilly, C., Sharma, S., Gray, D. K., Hampton, S. E., Read, J. S., Rowle,y R. J., Schneider, P., Lenters, J. D., McIntyre, P.B., Kraemer, B. M., Weyhenmeyer, G. A., Straile, D., Dong, B., Adrian, R., Allan, M. G., Anneville, O., Arvola, L., Austin, J., Bailey, J. L., Baron, J. S., Brookes, J. D., de Eyto, E., Dokulil, M. T.,

Hamilton, D. P., Havens, K., Hetherington, A. L., Higgins, S. N., Hook, S., Izmest'eva, L. R., Joehnk, K. D., Kangur, K., Kasprzal, P., Kumagai, M., Kuusisto, E., Leshkevich, 20 G., Livingtone, D. M., McIntyre, S., May, L., Melack, J. M., Mueller-Navarra, D. C, Naumenko, M., Noges, P., Noges, T., North, R. P., Plisnier, P. D., Rigosi, A., Rimmer, A., Rogora, M., Rudstam, L. G., Rusak, J. A., Salmaso, N., Samal, N. R., Schindler, D. E., Schladow, S. G., Schmid, M., Schmidt, S. R., Silow, E., Soylu, M. E., Teubner, K., Verburg, P., Voutilainen, A., Watkinson, A., Wiliamson, C. E., and Zhang G.: Rapid and highly variable warming of lake surface waters around the globe, Geophys. Res. Lett., 42, 10773–10781, https://doi.org/10.1002/2015GL066235, 2015.

Schmidt, W.: Über Temperatur und Stabilitätsverhaltnisse von Seen, Geogr. Ann., 10, 145–177, 1928.

Shatwell, T., Thiery, W., and Kirillin, G.: Future projections of temperature and mixing regime of European temperate lakes, Hydrol. Earth Syst. Sci., 23, 1533–1551, https://doi.org/10.5194/hess–23–1533–2019, 2019.

Woolway, R.I., and Merchant, C.J.: Worldwide alteration of lake mixing regimes in response to climate change, Nature Geoscience, 12, 271–276, https://doi.org/10.1038/s41561–019–0322–x, 2019.

---

## Author Response (AR1)

*General comments*

The manuscript entitled "Simulations of future changes in thermal structure of Lake Erken: Proof of concept for ISIMIP2b lake sector local simulation strategy" aims to assess the impacts global warming on the thermal characteristics of Lake Erken. Since future projections of global warming are at a daily time step, the authors first analyses the need to disaggregate the input variables to the hourly time step. The manuscript is well written. The topic is scientifically relevant.

**Response:** We thank the Referee 1 for the positive comments about the text. The paper was edited very carefully and modifications and improvements were made. Below, we address every comment and explain the corresponding changes in the manuscript.

*Specific comments*

Line 70-71: "It is the lake's relatively shallow depth and large surface area, which leads to large inter-annual variability in the timing and patterns of thermal stratification." Why is this? Perhaps explain in one or two sentences how this works and why this is different for relatively deep lakes or lakes with a small surface area.

**Response:** Mixing and stratification change in response to lake morphometry. Shallow lakes have lower heat storage, responding more directly to short-term variations in the weather conditions and heat can be transferred through the water column by wind mixing (Magee and Wu, 2017). However, deep lakes required greater wind speeds to complete the mix. Large surface areas or fetch increase the effects of mixing and vertical transfer of heat to the bottom (Rueda and Schladow, 2009).

**Changes in manuscript:** P19 L74-76.

Line 138-139: "More detailed description of the GRNN methods and models are given in the supplementary material to this paper." I was hoping to find equations on how the GRNN model calculates hourly estimations based on daily input, however, I could not find a detailed description of the GRNN methods in the supplementary materials.

**Response:** GRNN description was added in the supplement material section S1.

**Changes in supplement:** Section S1.

Line 158: "Schmidt stability", perhaps give a definition or equation of the Schmidt stability.

**Response:** The following Schmidt stability definition was added: resistance to mechanical mixing due to the potential energy inherent in the density stratification of the water column (Schmidt, 1928; Idso, 1973).

**Changes in manuscript:** P22 L190-191.

Line 170-174: "Air temperature, short-wave radiation, relative humidity and wind speed were temporarily disaggregated into hourly values from mean daily data, using the GRNN models. A database was constructed using 8 years of measurements. From this whole set of data, the first 5-years of data, that is, from 2008 to 2012, were used for

training, and 3-years of data from 2013 to 2015 were used for validating the results obtained." This sentence was confusing. After reading the methods section I first assumed this was about the calibration/validation of GOTM. However, later I realized it was about the calibration/validation of GRNN. I would expect these sentences in the methods section. Moreover, it would be good to mention clearly that there are two types of calibration/validation: that

40 of GOTM and that of GRNN.

**Response**: It has been moved from Results section 3.1. Hourly meteorological modelling to Material and Methods section 2.5. Temporal disaggregation of meteorological forcing data.

**Changes in manuscript:** P21 L147-154.

45 Line 192-193: "Temperature simulations for the validation period were more accurate (average RMSE of 0.66 ºC and NSE of 0.97) than for the calibration period (average RMSE of 0.95 ºC and NSE of 0.94), but in both periods the model performance was considered acceptable." I would expect that the validation period would be less accurate than the calibration period. Therefore, my first thought was then that perhaps the legend was swapped between calibration and validation. Yet, the authors later mention that this is "due the higher variability in observed water temperature

50 during the long calibration period." (Line 284-285). Then the question raises, which data set is more representative? Was the high variability during the calibration period actually quite normal and the validation period exceptionally uniform? And what does this mean for the validity of the output?

**Response:** Water temperature simulations were apparently more accurate for the validation period (2015-2016) than for the calibration period (2006-2014), which may appear unusual, but is due to the higher variability in observed

55 water temperature during the longer calibration period. Years with a longer duration of stratification and stronger stability, generally had higher simulation errors. Half of the eight-year calibration period exhibited these conditions, while the two-years used for validation both exhibited shorter duration of stratification and weaker stability.

| | year | RMSE (ºC) | | | thermal stratification | | | Schmidt stability (J m$^{-2}$) |
|---|---|---|---|---|---|---|---|---|
| | | 24h met | 1h met | synthetic 1h met | duration (days) | onset | loss | |
| Calibration | 2007 | 0.58 | 0.59 | 0.83 | 23 | 176 | 230 | 17.42 |
| | **2008** | **1.42** | **1.13** | **1.04** | **103** | **124** | **227** | **31.52** |
| | 2009 | 0.75 | 0.68 | 0.63 | 69 | 122 | 242 | 35.17 |
| | **2010** | **1.10** | **0.92** | **0.99** | **111** | **139** | **254** | **80.77** |
| | **2011** | **0.92** | **0.79** | **0.81** | **90** | **152** | **252** | **43.77** |
| | 2012 | 0.71 | 0.66 | 0.77 | 38 | 141 | 244 | 32.98 |
| | **2013** | **1.42** | **1.52** | **1.08** | **124** | **129** | **259** | **79.48** |
| | 2014 | 0.83 | 0.73 | 0.79 | 55 | 137 | 263 | 52.40 |
| Validation | 2015 | 0.59 | 0.66 | 0.65 | 71 | 162 | 240 | 17.60 |
| | 2016 | 0.69 | 0.73 | 0.71 | 67 | 173 | 239 | 47.25 |

60 **Changes in manuscript:** P30 L432-437.

Line 202: "As would be expected the simulations of bottom temperature were slightly less accurate" Why would this be expected?

**Response:** Higher errors were found at the lowest depth (15 m depth), part of this might have been caused by the presence of internal seiches in lake Erken which cannot be reproduced by 1D models such as GOTM.

Line 349-351: "Combined these results suggest important changes in the factors affecting lake biogeochemistry directly through changes in temperature and indirectly by influencing the availability of light and nutrients." The presented results only indicate an increase in temperature and stratification period. Since the presented data does not show how this affects biogeochemistry and the availability of light and nutrients, could the authors be a bit more specific on this in the conclusion? How do the authors think/speculate it will change (perhaps refer to the introduction where a short explanation is already given)?

**Response:** As mentioned in the introduction the projected changes in thermal stratification can influence many aspects of the lake ecosystem. Increases in thermal stability and duration of stratification can intensify hypolimnetic oxygen depletion (Foley et al., 2012; Schwefel et al., 2016) and hence induce enhanced internal phosphorous loading (North et al., 2014), increase the release of dissolved iron and manganese from sediments (Schultze et al., 2017) and also increase methane emissions (Grasset et al., 2018). Warming lake temperature affects biological rates of metabolism, growth and reproduction (Rall et al., 2012) and can promote cyanobacterial blooms (Paerl and Paul, 2012). When coupled to a *reduction* in *oxygen*-rich water, warming water temperature leads to a *lower fish* populations (O'Reilly et al., 2003; Yankova et al., 2017). Increase in evaporation associated with warming can lead to declines in lake water level (Hanrahan et al., 2010) with implications for water security. So these changes are expected in Lake Erken.
The expected changes in the lake ecosystem caused by changes in thermal stratification have been moved from 1. Introduction to 4. Discussion section and so our conclusions can be more understandable.
**Changes in manuscript:** P34-35 L580-587.

*Technical corrections*

Figures in general; 1) it would be good to have comparable axes per figure. For example, figure 3a has a y-axis going from 0-1.2 ºC, while figure 3b goes from 0-0.8 ºC. I suggest that the authors uniform the axis and perhaps use the normalized RMSE to compare the different subfigures 2).From the figures caption, it is not always clear if the predicted output is with GOTM or with GRNN. Perhaps include this information in the figure's caption. General: sometimes I read "wind_factor" and sometimes "wind factor" without "_". Is there a difference in meaning?
**Response:**
Figure 3 has been removed because GOTM model performance had been shown in twice (Figure 3 and Table 4).
GRNN and GOTM has been added to the figure captions to indicate if models have been used to disaggregate meteorological forcing data or to simulated water temperature.
Wind factor is the meaning of the parameter wind_factor.

Line 246-247: "Simulated changes were generally slight less for the simulations driven by daily forcing data as shown by the figures in parentheses". Put a dot after parentheses and change "slight" to "slightly"

**Response:** Change made.

100

Line 284: "were more accurate than for the calibration period (2006-2014) due the higher variability in observed water temperature" add "to" after "due".

**Response:** change made.

105 Figure 2: 1) the caption says that validation is figure 2a, 2c, 2e and 2g, however, the title of the figures suggest that validation is figure 2b, 2d, 2f and 2h. This is confusing. 2) Perhaps include the words "observations", "daily data", "hourly data", "synthetic hourly data" on the left side of/ or under the figure. It is now quite a puzzle to find which subfigure tells what. 3) Perhaps also include a difference graph where the difference between "observations" and respectively "daily data", "hourly data", "synthetic hourly data" is shown. From figure 2, it is now hard to see the

110 differences. (The same holds for figure 4, where it is hard to see the differences between historical and the rcp's)

**Response:**

Figure 2 has been renumbered and a subtitle added to each subfigure. **Changes in manuscript:** P54.

Figure S7 has been added to the supplementary material showing the differences between simulated (when the lake model was forced with daily, hourly and synthetic hourly meteorological forcing data) and observed water

115 temperature. **Changes in supplement:** Section S2.

Figure S14 has been added to the supplementary material showing the differences between the historical and RCP 2.6 scenarios, and the historical and RCP 6.0 scenarios for the IPSL-CM5A-LR projection (when the lake model was forced at daily resolutions). **Changes in supplement:** Section S4.

120 Figure 5 and 6: In figure 5i, the authors indicate the words "deeper" and "shallower" with arrows. This really increases the readability of that specific subfigure and the same would help the reader in all other subfigures.

**Response**: In Figures 6 and 7 (in the latest version of the manuscript), the arrow and the words "deeper" and "shallower" have been removed for easy viewing of the figure. However, in the figure caption was added the meaning of values greater or less than 0 of each of the thermal indices: changes in thermal metrics greater than 0 show an

125 increase and lower than 0 show a decrease.

**Changes in manuscript:** P62-63.

**Changes in supplement:** Sections S3-S4.

**Anonymous Referee 2**

*General comments*

The article "Simulations of future changes in thermal structure of Lake Erken: Proof of concept for ISIMIP2b lake sector local simulation strategy" presents impacts of changing climate on lake water temperature. The article is of very high interest and very well written, the work is thoroughly executed and discussion is relevant. The authors used a hydrodynamic lake model GOTM with 4 GCM/RCMs, using Generalized Regression Artificial Neural Network to disaggregate daily climate into hourly data. The GOTM model was able to reproduce observed lake temperature data for current time period (8 years). The model was then executed with climate forcing data from 4 GCM/RCMs.

**Response:** We thank the Referee 2 for the positive comments about the text. The paper was edited very carefully and modifications and improvements were made. Below, we address every comment and explain the corresponding changes in the manuscript.

*Specific comments*

I would recommend expanding on the Methods section to provide more information that is critical in understanding the study, its aims, and results. It is unclear why the authors chose to consider 2006-2099 as the future even though the period begins 13 years ago. It is also unclear why this full period is evaluated without any consideration of the changes that occur from 2006 to 2099 based on the trend analyses also included in the manuscript. It seems that changes that can occur during this "future" period are considered representative of changes that will occur by 2099. The averages from this 94-year period are compared to averages from a 30-year period of 1975-2005. The variability during a 30-year period and during a 94-year period with a significant trend is expected to differ and this affects the projected changes.

A more typical approach in many climate impact studies is to select two 30-year periods, one that represents a current climate (reference period, e.g. 1981-2010) and one that represents a future climate (e.g., mid-century 2041-2010 or late century 2071-2100). Forcing data from the same climate model would then be used as model inputs for both time periods; the difference between these results would represent the projected impact. It is also not clear from the manuscript how were the reference period values calculated for calculation of anomalies from the respective GCM/RCMs during the reference time period.

The results for the mid-century and late century should be added to the manuscript to evaluate how the change progresses; alternatively, the current results can be replaced with the late century period as that seem to be the focus of the "proof of concept" study.

It is also important to include information on the variability of the simulated thermal indices due to the climate model selection, i.e. present information for all 4 GCM/RCMs for the reference time period. That can give indication to the significance of the projected impact.

**Response:**

Climate impact studies can be approached in two ways: (1) assessing the difference in mean lake conditions (for example, mean surface temperature) between the reference periods and both mid-century and late-century (Woolway and Merchant, 2019) or (2) long-term trend analysis (O'Reilly et al., 2015; Shatwell et al, 2019; Moras et al., 2019). The use of anomalies or absolute values in trend analysis does not change the value of the slope. The use of anomalies in frequency distribution figures provide an alternative method of comparing the changes simulated by the future climate scenarios. So we consider it appropriate to combine both approaches. We have also added information about the reference period and the same analysis has been made to the meteorological variables in order to understand the variability of the projected thermal metrics derived from GCMs.

**Changes in manuscript:** Material and methods: section 2.8 Statistical analysis P25 L255-278, Results: section 3.3 Climate data projections P27 L323-342, section 3.4 Long-term modelled changes in thermal stratification P27-30 L343-4219 and section 3.5 Comparison between long-term thermal metrics derived from daily and hourly climate data P30 L420-427. Discussion: P32-34 L499-579.

**Changes in supplement:** Sections S3-S5.

The information on the GOTM model for Lake Erken is very limited; the methods section should be expanded to include more details on the model structure, e.g. vertical resolution, inflow and outflow from the lake, etc.

**Response:** The GOTM model version 5.1 was used in this study. The meteorological parameters for running the model were air temperature (ºC), wind speed (m s$^{-1}$), short-wave radiation (W m$^{-2}$), cloud cover (dimensionless, 0-1), relative humidity (%), atmospheric pressure (hPa) and precipitation (mm day$^{-1}$or mm hour$^{-1}$). Inflows and outflows were not included in this study, and water level was considered fixed in the simulations. This version of GOTM did not have the ability to simulate lake ice, so for this study the inverse stratification period was not analysed. Moras et al., (2019), has shown that despite this limitation, the mode is able to accurately simulate water temperature and the phenology of thermal stratification during the remainder of the year. The initial conditions for water temperature were derived from a measured vertical profile. GOTM was run at hourly model computational time step, and simulated water temperature was saved as daily mean values each 0.5 m (42 layers).

**Changes in manuscript:** P21-22 L155-163.

Some relevant parts should be moved from Results to Methods (e.g., the beginning of section 3.1 and 3.3. Also, the periods used in the calibration (training) & validation periods for GOTM and GRNN should be put into the context between these two models. It is not readily apparent from the manuscript.

**Response:**

The beginning of sections 3.1 and 3.3 have been moved to 2. Materials and Methods section 2.5 Temporal disaggregation of meteorological forcing data 2.8 Statistical analysis respectively.

The GRNN training and validation periods do not fit into GOTM calibration periods. Putting these periods in context does not produce significant changes in the GRNN models performance (see table below) but it would entail a high computational cost since changing the GRNN models would require all the GCM scenarios to be disaggregated a second time and all GCM scenarios to be run again using these alternative data.

70

| | Air temperature (ºC) | | | Relative humidity (%) | | | Wind speed (m s$^{-1}$) | | | Short-wave rad (W m$^{-2}$) | | |
|---|---|---|---|---|---|---|---|---|---|---|---|---|
| | BIAS | RMSE | NSE | BIAS | RMSE | NSE | BIAS | RMSE | NSE | BIAS | RMSE | NSE |
| Training: 2006-2014 | 0.00 | 0.32 | 1.00 | 0.00 | 0.96 | 1.00 | -0.01 | 1.15 | 0.74 | 0.00 | 8.39 | 1.00 |
| Validation: 2015-2016 | 0.03 | 0.70 | 0.95 | 0.44 | 2.09 | 0.69 | -0.07 | 2.50 | 0.60 | 0.08 | 18.15 | 0.86 |
| Training: 2008-2012 | 0.00 | 0.26 | 1.00 | 0.00 | 0.79 | 1.00 | -0.01 | 1.06 | 0.78 | 0.00 | 6.35 | 1.00 |
| Validation: 2013-2015 | -0.06 | 0.32 | 0.94 | 0.34 | 1.02 | 0.69 | -0.01 | 1.37 | 0.58 | -0.04 | 8.20 | 0.87 |

**Changes in manuscript:** P21 L147-154 and P25 L255-263.

It should be emphasized that training the temperature disaggregation algorithm on the current diurnal patterns means those
75 current patterns will be projected to the future time series and any potential changes in diurnal pattern from the changing climate are ignored.

**Response:** GRNNs proved to be an effective method to disaggregate daily GCM forcing to an hourly temporal resolution for different weather variables such as air temperature, short-wave radiation, etc. However, GRNNs require a training phase, in which the diurnal patterns to be learned are presented to the network from historical meteorological measurements, and
80 therefore if there are future changes in diurnal patterns, these cannot be reproduced. In addition, there is a high computational cost of disaggregating and storing the long-term daily climate data into an hourly data set.

**Changes in manuscript:** P33 L539-544.

*Technical comments*

Increases are given to 0.01 ºC – what is the accuracy of the measurement and of the simulations? Is this accuracy adequate?
85 **Response**: The accuracy of thermocouple sensor is approximately ± 0.1 ºC and can at times be somewhat better than 0.1. The simulated water temperature is given with 7 decimals. So two decimal places in the GOTM model performance are adequate, and match the best expected performance of our monitoring data.

90   L 68 –dimictic?

    **Response:** Change made.

    L89 Mean sea level (,) pressure (,) relative humidity and precipitation were measured – missing commas?

    **Response:** Change made.

95   Section 2.6. It would be useful to include model performance for other thermal indices used for evaluation of change, e.g. duration of thermal stratification

    Response:

    **Response:** The model performance for the duration, onset and loss of stratification has been added to the section 3.2 Lake Model performance (Table 4).

100   **Changes in manuscript:** P26-27 L 318-322.

    L 162: Schmidt's stability – needs a reference/ brief explanation

    **Response:** The following Schmidt stability definition was added: resistance to mechanical mixing due to the potential energy inherent in the density stratification of the water column (Schmidt, 1928; Idso, 1973).

    **Changes in manuscript:** P6 L190-191 and P44.

105

    L 231: this model handicap and any other should be described in section 3.2

    **Response**: The GOTM model version 5.1 did not have the ability to simulate lake ice, so for this study the inverse stratification period was not analysed. Moras et al., (2019), has shown that despite this limitation, the mode is able to accurately simulate water temperature and the phenology of thermal stratification during the remainder of the year. A new GOTM model version

110   5.4 with ice-module was released after this project was submitted, allowing to evaluate the effect of the lack of ice module on the onset of the direct stratification. The onset of direct stratification was derived from simulations of water temperature with GOTM version 5.1 and 5.4 from 2006 to 2016. The RMSE between the onset of direct stratification from GOTM version 5.1 and 5.4 was 5.22 days showing a slight impact the lack of ice-module on the onset of the direct stratification.

| onset of direct stratification | |
| --- | --- |
| GOTM v5.1 | GOTM v5.4 |
| 2007-04-27 | 2007-04-16 |
| 2008-04-27 | 2008-04-26 |
| 2009-04-27 | 2009-04-27 |
| 2010-05-01 | 2010-05-13 |
| 2011-04-24 | 2011-04-25 |
| 2012-05-03 | 2012-05-01 |

| | |
|---|---|
| 2013-05-09 | 2013-05-09 |
| 2014-04-21 | 2014-04-21 |
| 2015-05-22 | 2015-05-22 |
| 2016-05-04 | 2016-05-03 |

115

**Changes in manuscript:** P22 L159-162.

L 312-314 However, the dominant wind (is) along : : : missing word?

**Response:** Change made.

120 L 322, 324 When GOTM was forcing with : : : forced?

**Response:** Change made.

L 350: it would be good to put the statement into context; what kind of changes can be expected with these increases in temperature?

**Response:** The expected changes in the lake ecosystem caused by an increase in water temperature have been moved from 1.

125 Introduction to 4. Discussion section.

**Changes in manuscript:** P34-35 L580-587.

Figure 2 heading: figure shows calibration as plots a, c, e, and g, but the caption says these are validations. I would recommend including a similar plot but with model residuals (perhaps in Supplementary materials); that would make any

130 differences much easier to see especially on the timing.

**Response:**

Figure 2 has been renumbered and a subtitle added to each subfigure. **Changes in manuscript:** P54.

Figure S7 has been added to the supplementary material showing the differences between simulated (when the lake model was forced with daily, hourly and synthetic hourly meteorological forcing data) and observed water temperature. **Changes in**

135 **supplement:** Section S2.

Figure 3: it would be helpful if the scale on y axis with the same units had the same range (a-d)

Figure 3 has been removed because GOTM model performance had been shown in twice (Figure 3 and Table 4).

**Anonymous Referee 3**

*General comments*

The manuscript entitled "Simulations of future changes in thermal structure of Lake Erken: Proof of concept for ISIMIP2b lake sector local simulation strategy" showed the effects of different time-scale forcing data and 4 model forcing and also the 2 RCP future scenario on the simulation with GOTM lake model over Lake Erken. It projected the similar future changing trends of thermal conditions and is helpful for local to understand the effects of climate change and adapt it.

**Response:** We thank the Referee 3 for the positive comments about the text. The paper was edited very carefully and modifications and improvements were made. Below, we address every comment and explain the corresponding changes in the manuscript.

*Specific comments*

The work focused on daily characteristics of future thermal contracture in Figure 4-6. The simulated future changing trends are mostly similar with hourly or daily forcing. But lots of work were done to compare the simulation results with different historical data which may be simplified or removed. Then the work could pay more attention to the future changing characteristics.

**Response:** The purpose of this paper is twofold: (1) evaluate the importance of diurnal forcing in 1D lake model and (2) assess the long-term impacts of climate change on the thermal structure of Lake Erken. Therefore, we do not consider it appropriate to simplify or remove the first purpose. The difference in mean lake conditions between the reference periods and both mid-century and late-century and long-term trend analysis has been analysed for the climate data and thermal metrics. And also the differences of each meteorological variable and thermal metric were evaluated when the lake model was forced at daily and hourly resolutions respectively.

**Changes in manuscript:** Material and methods: section 2.8 Statistical analysis P25 L255-278, Results: section 3.3 Climate data projections P27 L323-342, section 3.4 Long-term modelled changes in thermal stratification P27-30 L343-4219 and section 3.5 Comparison between long-term thermal metrics derived from daily and hourly climate data P30 L420-427. Discussion: P32-34 L499-579.

**Changes in supplement:** Sections S3-S5.

L244-246 "Rates of change in whole-lake temperature calculated for over the length for RCP2.6 and 6.0 scenarios were projected to 245 increase except in the case of GFDL-ESM2M which showed weaker or non-significant changes for all measures of thermal stratification." did not match with Table 5.

**Response:** We do not agree with this comment. Table S8 and Table 5 show the trend analysis under RCP 2.6 and 6.0 respectively for the period 2011-2100. For RCP 2.6 the whole-lake temperature projected under GFDL-ESM2M shows a nonsignificant increase, and for RCP 6.0 the project increase associated with GFDL-ESM2M was the lowest of the GCMs. For RCP 6.0 the increase in whole-lake temperature ranged from 0.26 to 0.14 °C decade$^{-1}$.

35    Some parts were hardly understood, such as "For RCP 6.0, the projected rate of change ranged from 0.15 to 0.27 °C decade-1 (0.11 to 0.19 °C decade-1). IPSLCM5A- LR projected the largest increase being 0.59 °C (0.43 °C) under RCP 2.6 °C and 2.51 °C (1.79 °C) under RCP 6.0". And IPSL-CM5A-LR did not project the largest temperature increase under RCP 2.6 as showed in Table 5.

**Response:** We totally agree, sometimes it's hard to understand. The results have been rewritten, reducing the large amount of
40    numbers in the text, making it more readable. All the results can be found in the Figures and Tables of both the manuscript and the supplement material. IPSL-CM5A-LR did not project the largest temperature increase under RCP 2.6, under scenario future RCP 2.6 HadGEM2-ES projected the largest increase in surface temperature, being 0.15 °C decade$^{-1}$. The trend analysis has been carefully reviewed and the results rewritten.

**Changes in manuscript:** P25-27 L281-322, P30-32 L429-498.

Because the lake model parameters are different for different forcing in Table 2. It's hard to know the source of the simulation difference in Table 4 and to evaluate the effects of the time-scale of forcing.

**Response:** One of the purposes of this study was to test the ability of a 1D lake model (GOTM) to simulate daily water temperature using daily vs hourly meteorological data, i.e. evaluate the importance of diurnal forcing in 1D lake model. In all
50    cases the lake model was ran at hourly model computational time step when the meteorological forcing was provided at either daily or hourly frequencies. In each case a separate calibration was run using the same observed data for comparison, simulated output derived from the models forced at daily and hourly resolution. We felt that this was the fairest and most representative way to test how the model would actually be applied with the different forcing data. When GOTM was forced at daily resolutions, there is no diurnal variability in the input, which leads to changes in heat fluxes. However it became apparent that
55    variations in model parameters resulting from the different calibrations compensated for some of the differences between observations and simulations based on the different time-scale of forcing. We now point this out more clearly in the paper.

**Changes in the manuscript:** P33 L529-539.

L230 "From these average yearly values were calculated using the months between April and September, due to the fact that
60    the GOTM model was not able to simulate lake ice and winter water temperatures at the same level of accuracy as during the remainder of the year". Does the inaccurate simulation of lake temperature in winter affect the temperature simulation without ice? L68 "The lake is dimictic with summer stratification usually occurring beginning in May-June and ending in August-September, while ice cover occurs from December-February to April-May." Why the average yearly values were calculated including April?

65   **Response**: The GOTM model version 5.1 did not have the ability to simulate lake ice, so for this study the inverse stratification period was not analysed. Moras et al., (2019), has shown that despite this limitation, the mode is able to accurately simulate water temperature and the phenology of thermal stratification during the remainder of the year. A new GOTM model version 5.4 with ice-module was released after this project was submitted, allowing to evaluate the effect of the lack of ice module on the onset of the direct stratification. The onset of direct stratification was derived from simulations of water temperature with

70   GOTM version 5.1 and 5.4 from 2006 to 2016. The RMSE between the onset of direct stratification from GOTM version 5.1 and 5.4 was 5.22 days showing a slight impact the lack of ice-module on the onset of the direct stratification.

| onset of direct stratification | |
|---|---|
| GOTM v5.1 | GOTM v5.4 |
| 2007-04-27 | 2007-04-16 |
| 2008-04-27 | 2008-04-26 |
| 2009-04-27 | 2009-04-27 |
| 2010-05-01 | 2010-05-13 |
| 2011-04-24 | 2011-04-25 |
| 2012-05-03 | 2012-05-01 |
| 2013-05-09 | 2013-05-09 |
| 2014-04-21 | 2014-04-21 |
| 2015-05-22 | 2015-05-22 |
| 2016-05-04 | 2016-05-03 |

Annual ice cover observations of the onset and loss of ice cover made at lake Erken since 1941 (Moras et al., 2019) showed a

75   decreased since 1941 by 7.34 day decade$^{-1}$ (57 days from 1941 to 2017), consistent with changes in air temperature. For this reason, we consider relevant in our long-term study to include April in our analysis.

The manuscript was submitted in 2019. It's confused to compare 2006-2099 with 1975-2005 to get the future change.

  **Response:** we totally agree, the choice of reference period is always controversial because the projected impact depends on it.

80   Initially we used as a reference period the last 30 years of the historical scenario (1975-2005) for each GCM, since from 2006 they were already future projections. However, we have decided to slightly update our reference period to 1981-2010. The table shows the trend analysis for the period 2006-2100 relative to 1975-2005 and for the period 2011-2100 relative to 1981-2010 for HadGEM2-ES under RCP 6.0. The differences are almost unnoticeable, so we do not consider it necessary to update our reference period to 1990-2019.

85

|  | HadGEM2-ES RCP 6.0 | | | |
|  | reference period: 1975-2005 | | reference period: 1981-2010 | |
|  | 24h met | 1h met | 24h met | 1h met |
|---|---|---|---|---|
| air temperature (°C) | 0.44 °C dec$^{-1}$ | 0.33 °C dec$^{-1}$ | 0.43 °C dec$^{-1}$ | 0.32 °C dec$^{-1}$ |
| surface temperature (°C) | 0.38 °C dec$^{-1}$ | 0.28 °C dec$^{-1}$ | 0.38 °C de-1 | 0.27 °C dec$^{-1}$ |
| bottom temperature (°C) | 0.07 °C dec$^{-1}$ | ns | 0.06 °C dec-1 | ns |
| whole-lake temperature (°C) | 0.25 °C dec$^{-1}$ | 0.17 °C dec$^{-1}$ | 0.25 °C dec$^{-1}$ | 0.16 °C dec$^{-1}$ |
| Schmidt stability (J m$^{-2}$) | 7.79 J m$^{-2}$ dec$^{-1}$ | 6.22 J m$^{-2}$ dec$^{-1}$ | 7.97 J m$^{-2}$ dec$^{-1}$ | 6.50 J m$^{-2}$ dec$^{-1}$ |
| thermocline depth (m) | 0.12 m dec$^{-1}$ | 0.12 m dec-$^{1}$ | 0.13 m dec$^{-1}$ | 0.13 m dec$^{-1}$ |

Does the lake model need downward longwave radiation drive? What's the usage of the cloud cover when there is the downward shortwave radiation?

GOTM internally calculates net long-wave radiation from cloud cover according to Clark et al. (1974). Cloud cover for long-term water temperature simulations was estimated from bias-corrected model data according to Martin and McCutcheon (1999):

$$H_{SW} = H_0 \cdot a_t \cdot (1 - R_s) \cdot C_a$$

where $H_{sw}$ is the short-wave solar radiation (W · m$^{-2}$), $H_0$ is the amount of radiation reaching the earth's outer atmosphere (W · m$^{-2}$), $a_t$ is an atmospheric transmission term, $R_s$ albedo or reflection coefficient, and $C_a$ is the fraction of solar radiation not absorbed by clouds.

$$C_a = 1 - 0.65 \cdot C_l^2$$

where $C_l$ is the fraction of the sky covered by clouds.

Cloud cover would be:

$$C_l = \sqrt{\frac{1 - \frac{H_{SW}}{H_0 \cdot a_t \cdot (1 - -R_s)}}{0.65}}$$

Usually the simulation in the calibration period is better. Why temperature simulations in the validation period were more accurate in the manuscript?

**Response:** Water temperature simulations were apparently more accurate for the validation period (2015-2016) than for the calibration period (2006-2014), which may appear unusual, but is due to the higher variability in observed water temperature during the longer calibration period. Years with a longer duration of stratification and stronger stability, generally had higher simulation errors. Half of the eight-year calibration period exhibited these conditions, while the two-years used for validation both exhibited shorter duration of stratification and weaker stability.

| | year | RMSE (ºC) | | | thermal stratification | | | Schmidt stability (J m$^{-2}$) |
|---|---|---|---|---|---|---|---|---|
| | | 24h met | 1h met | synthetic 1h met | duration (days) | onset | loss | |
| Calibration | 2007 | 0.58 | 0.59 | 0.83 | 23 | 176 | 230 | 17.42 |
| | **2008** | **1.42** | **1.13** | **1.04** | **103** | **124** | **227** | **31.52** |
| | 2009 | 0.75 | 0.68 | 0.63 | 69 | 122 | 242 | 35.17 |
| | **2010** | **1.10** | **0.92** | **0.99** | **111** | **139** | **254** | **80.77** |
| | **2011** | **0.92** | **0.79** | **0.81** | **90** | **152** | **252** | **43.77** |
| | 2012 | 0.71 | 0.66 | 0.77 | 38 | 141 | 244 | 32.98 |
| | **2013** | **1.42** | **1.52** | **1.08** | **124** | **129** | **259** | **79.48** |
| | 2014 | 0.83 | 0.73 | 0.79 | 55 | 137 | 263 | 52.40 |
| Validation | 2015 | 0.59 | 0.66 | 0.65 | 71 | 162 | 240 | 17.60 |
| | 2016 | 0.69 | 0.73 | 0.71 | 67 | 173 | 239 | 47.25 |

110

**Changes in manuscript:** P30 L432-437.

L 110 "under four emission scenarios". As shown in the manuscript, there were only 2 emission scenarios.

**Response:** Change made.

115

If the years for calibration and validation match the years for training and validating, it may be better.

The GRNN training and validation periods do not fit into GOTM calibration periods. Putting these periods in context does not produce significant changes in the GRNN models performance (see table below) but it would entail a high computational cost since changing the GRNN models would require all the GCM scenarios to be disaggregated a second time and all GCM

120 scenarios to be run again using these alternative data.

| | Air temperature (ºC) | | | Relative humidity (%) | | | Wind speed (m s$^{-1}$) | | | Short-wave rad (W m$^{-2}$) | | |
|---|---|---|---|---|---|---|---|---|---|---|---|---|
| | BIAS | RMSE | NSE | BIAS | RMSE | NSE | BIAS | RMSE | NSE | BIAS | RMSE | NSE |
| Training: 2006-2014 | 0.00 | 0.32 | 1.00 | 0.00 | 0.96 | 1.00 | -0.01 | 1.15 | 0.74 | 0.00 | 8.39 | 1.00 |
| Validation: 2015-2016 | 0.03 | 0.70 | 0.95 | 0.44 | 2.09 | 0.69 | -0.07 | 2.50 | 0.60 | 0.08 | 18.15 | 0.86 |
| Training: 2008-2012 | 0.00 | 0.26 | 1.00 | 0.00 | 0.79 | 1.00 | -0.01 | 1.06 | 0.78 | 0.00 | 6.35 | 1.00 |
| Validation: 2013-2015 | -0.06 | 0.32 | 0.94 | 0.34 | 1.02 | 0.69 | -0.01 | 1.37 | 0.58 | -0.04 | 8.20 | 0.87 |

[revised manuscript text omitted]
 MBE | 24h met RMSE | 24h met NSE | 1h met MBE | 1h met RMSE | 1h met NSE | synthetic 1h met MBE | synthetic 1h met RMSE | synthetic 1h met NSE |
|---|---|---|---|---|---|---|---|---|---|
| full-profile temp (ºC) | -0.08 | 1.04 | 0.93 | -0.02 | 0.94 | 0.94 | -0.02 | 0.88 | 0.95 |
| surface temp (ºC) | -0.04 | 0.69 | 0.97 | 0.04 | 0.72 | 0.97 | -0.01 | 0.61 | 0.98 |
| bottom temp (ºC) | -0.06 | 1.33 | 0.83 | 0.07 | 1.24 | 0.85 | -0.11 | 1.16 | 0.87 |
| whole lake temp (ºC) | -0.07 | 0.57 | 0.98 | -0.03 | 0.52 | 0.98 | -0.01 | 0.49 | 0.98 |
| Schmidt stability (J m$^{-2}$) | 0.53 | 22.09 | 0.85 | 0.59 | 21.69 | 0.85 | 0.76 | 19.64 | 0.88 |
| thermocline depth (m) | 0.58 | 2.77 | 0.32 | 0.84 | 3.07 | 0.22 | 0.43 | 2.84 | 0.32 |
|  | validation 24h met MBE | 24h met RMSE | 24h met NSE | 1h met MBE | 1h met RMSE | 1h met NSE | synthetic 1h met MBE | synthetic 1h met RMSE | synthetic 1h met NSE |

[revised manuscript text omitted]

CALIBRATION

(1a) observations

(1b) simulations driven by 24h met

(1c) simulations driven by 1h met

(1d) simulations driven by synthetic 1h met

VALIDATION

(2a) observations

(2b) simulations driven by 24h met

(2c) simulations driven by 1h met

(2d) simulations driven by synthetic 1h met

840     **Figure 2. GOTM water temperature simulations. Daily averaged water temperature in Lake Erken for the**  **calibration (****1a)-(1b)-(1c)-(1d) and validation (2a)-(2b)-(2c)-(2d) periods: observations (****1a)-(2a), simulations driven by daily meteorological data (****1b)-(2b), hourly meteorological data (****1c)-(2c) and synthetic hourly meteorological data (****1d)-(2d).**

[Figure]

845 Figure 3. GOTM model performance metrics for prediction of (a) full-profile temperature which compared simulated and measured data at all possible depths, (b) surface temperature, (c) bottom temperature, (d) whole-lake temperature, (e) Schmidt stability and (f) thermocline depth. The mean (horizontal line) is also shown

[Figure]

[Figure]

**Figure 3.**Figure 4. **Temperature isopleth diagrams for the (a) historical, (b) RCP 2.6 and (c) RCP 6.0 scenarios showing results from the lake model forced with daily IPSL-CM5A-LR projections. The temperature matrix used to make these plots was created by averaging the simulated daily temperature profiles for every year in each scenario.**

850

855    Evolution of annual average projected  anomalies  (from April to September)  for (1a)-(2a) whole-lake temperature, (1b)-(2b) surface temperature, (1c)-(2c) bottom temperature, (1d)-(2d) Schmidt stability and (1e)-(2e) thermocline depth  showing results when the lake model was forced with daily GFDL-ESM2M, HadGEM2-ES, IPSL-CM5A-LR and MIROC5 projections.  from 2011 to 2100 under RCP 2.6 and 6.0. Anomalies are relative to reference period (1981-2010).

[Figure]

860

[Figure]

Figure 5.

**Evolution of annual average projected anomalies (from April to September) for (1a)-(2a) duration,  1b)-(2b) onset and 1c)-(2c) loss of stratification  showing results when the lake model was forced with daily GFDL-ESM2M, HadGEM2-ES, IPSL-CM5A-LR and MIROC5 projections from 2011 to 2100 under RCP 2.6 and 6.0. Anomalies are relative to reference period (1981-2010).**

[Figure]

**Figure 6. ~ Changes in annually averaged thermal metrics (from April to September) (2a)-(3a) whole-lake temperature, (2b)-(3b) surface temperature, (2c)-(3c) bottom temperature, (2d)-(3d) Schmidt stability and (2e)-(3e) thermocline depth under RCP 6.0, showing results when the lake model was forced with daily GFDL-ESM2M, HadGEM2-ES, IPSL-CM5A-LR and MIROC5 projections. All changes are for mid-century (2041-2070) and late-century (2071-2100) are relative to reference period (1981-2011). The mean (vertical line) is also shown. Changes in thermal metrics greater than 0 show an increase and lower than 0 show a decrease.**

870

[Figure]

**Figure 7. Changes in annually averaged thermal metrics (from April to September) (2a)-(3a) duration, (2b)-(3b) onset and (2c)-(3c) loss of stratification under RCP 6.0, showing results when the lake model was forced with daily GFDL-ESM2M, HadGEM2-ES, IPSL-CM5A-LR and MIROC5 projections. All changes are for  mid-century (2041-2070) and late-century (2071-2100) are relative to reference period (1981-2011). The mean (vertical line) is also shown. Changes in thermal metrics greater than 0 show and increase and lower than 0 show a decrease.**

875